# The transforming dairy sector in Ethiopia

**Bart Minten** [1]*, **Yetimwork Habte**[2], **Seneshaw Tamru**[3], **Agajie Tesfaye**[4]

1 International Food Policy Research Institute, Addis Ababa, Ethiopia, 2 Policy Studies Institute, Addis Ababa, Ethiopia, 3 International Growth Center, Addis Ababa, Ethiopia, 4 Ethiopian Institute of Agricultural Research, Addis Ababa, Ethiopia

* b.minten@cgiar.org

## Abstract

In the transformation of agri-food systems in developing countries, we usually see rapid changes in the dairy sector. However, good data for understanding patterns and inclusiveness of this transformation are often lacking. This is important given implications for policy design and service and technology provision towards better performing dairy sectors in these settings. Relying on a combination of unique diverse large-scale datasets and methods, we analyze transformation patterns in the dairy value chain supplying Addis Ababa, the capital of Ethiopia, the second most populous country in Africa. Over the last decade, we note a rapid increase in expenditures on dairy products by urban consumers, especially among the better-off. Relatedly, the number of dairy processing firms in Ethiopia tripled over the same period, supplying a significant part of these dairy products, especially pasteurized milk, to the city's residents. Upstream at the production level, we find improved access to livestock services, higher adoption of cross-bred cows, an increase in milk yields, expanding liquid milk markets, a sizable urban farm sector supplying almost one-third of all liquid milk consumed in the city, and an upscaling process with larger commercial dairy farms becoming more prevalent. However, average milk yields are still low and not all dairy farmers are included in this transformation process. Small farms with dairy animals as well as those in more remote areas benefit less from access to services and adopt less these modern practices. For these more disadvantaged farmers, stagnation in milk yields and even declines–depending on the data source used–are observed.

## Introduction

Structural transformation of economies is an essential process for poverty reduction and welfare improvements [1, 2]. Labor and land productivity are expected to improve and the relative importance of high-value agriculture–including fruits and vegetables, fish, and livestock–increases, leading to significant changes in the demands made on agricultural production systems [3, 4]. To better understand ongoing transformation processes in developing countries, we look in this study at the case of Ethiopia, the second most populous country in Africa. Ethiopia has been one of the fastest growing economies in Africa in the last decade. In the ongoing transformation of Ethiopian food systems, animal source foods (ASF) play an increasingly important role. As in other growing economies, the relative importance of cereals in total

**Data Availability Statement:** Data: Habte, Yetimwork, 2019, "Dairy Value Chain Survey in Ethiopia: Producer Survey", https://doi.org/10.7910/DVN/7AIKZH, Harvard Dataverse, V1, UNF:6: t2Vb/WuOUY3a3+tb8BA4QA== [fileUNF].

**Funding:** The research presented here was conducted as part of the CGIAR Research Program on Policies, Institutions, and Markets (PIM), which is led by IFPRI. This study was made possible by the generous support of the American people through USAID under the Feed the Future Innovation Lab for Livestock Systems (LSIL), which is implemented by the Institute of Food and Agricultural Sciences of the University of Florida in partnership with the International Livestock Research Institute (ILRI). LSIL is funded by the United States Agency for International Development (USAID) through a five-year Leader with Associates Cooperative Agreement Award No. AID-OAA-L-15-00003. The funders had no role in study design, data collection and analysis, decision to publish, or preparation of the manuscript.

**Competing interests:** The authors have declared that no competing interests exist.

food expenditures is decreasing in Ethiopia, and we begin to see a shift toward more pre-ferred–but also more expensive–foods, including ASF [5].

The dairy sector is seen as an important high-value growth sector in the process of agricultural and economic transformation that, moreover, has the potential to provide good income opportunities for the poor [6–8]. Hence, understanding changes in the dairy sector is important given the likely high poverty alleviation, as well as nutritional, impacts growth in the sector might foster [9–12]. Despite its importance, however, it is not clear how any transformation in the dairy value chain in developing countries is unfolding.

In this paper, we fill this gap using unique data from Ethiopia. Consumption of dairy products is considerably higher in urban areas in Ethiopia. Thus, the commercial rural-urban value chain reflects the most important value chain–as a pull factor for more commercially oriented livestock production–in the country. We therefore focus, using a combination of administrative, quantitative, and qualitative data and different types of methods, on analyzing the changes that are happening in the dairy value chain that supplies the city of Addis Ababa, the capital and the biggest city in the country.

We further look at two important factors of inclusion in dairy transformation processes. First, we look at the issue of location and market access as measured by remoteness from Addis Ababa. Previous research has shown the important implications of market access on input use, profitability, and overall agricultural performance [13, 14]. A second important variable is farm size, as measured by the number of cows on the farm. Research in other settings has documented that transformation regularly excludes small farmers because of relatively higher coordination costs that downstream firms in the value chain incur in their commercial engagements with these small farms. The inclusion of smallholders in agricultural transformation, however, is deemed important for broad-scale welfare improvements [15].

We find important changes over the last decade. Downstream, we note an increase in the consumption of dairy products in general and of liquid milk in particular, especially by the better-off population. Midstream, we see a rapid rise in the number of dairy processing firms with a tripling in their number over the last decade. Upstream, we note increasing dairy supplies in both rural and suburban areas. Producers are increasingly selling to commercial dairy processing companies. However, we also observe the surprising importance of urban dairy farms that focus on informal non-pasteurized liquid milk markets. In rural areas that are well-connected with transport links to Addis Ababa, we see higher adoption of cross-bred cows and commercial feeds, and better access to services (animal health and extension), contributing to significantly higher milk yields per animal. In output markets, we observed that well-connected areas have shifted away from butter to the sales of liquid milk. We further note an upscaling process with larger dairy farms becoming more prevalent.

However, not all dairy farmers are included in this transformation process as small farms and farms in more remote areas participate disproportionally less. For the latter, a stagnation in milk yields over time or even a decline–depending on the data source used–is noted. Overall, we see improvements in modern input adoption and yields in the dairy sector, seemingly linked to incentives from increasing urban demand. But there are still many challenges, as milk yields overall remain low and consumer prices are high. Moreover, the exclusion of remote and small dairy farmers from participating in the unfolding transformation process is an important issue.

## Materials and methods

The study was reviewed and approved by the International Food Policy Research Institute's Institutional Review Board (IRB application approval number: 17-11-01; IRB #00007490; FWA #00005121). The study met the criteria for expedited review using survey procedures as

set forth in the code of federal regulations (45 CFR 46.110 Category 7) and presented no more than minimal risks to human subjects. Proper written consent requirements have also been met. In compliance with the IRB approval for work with human subjects, respondents could skip any question or withdraw from the study at any time.

Different data sources are used in this study. First, we rely on several secondary data sources. We obtained administrative data at the level of the woreda (district), zone, and region from the Ministry of Agriculture. We also obtained data from the Ethiopian Meat and Dairy Industry Development Institute. We further use consumption data from the nationally representative Household Income, Consumption and Expenditure Surveys (HICES) conducted by the Central Statistics Agency (CSA), particularly data from the 2004/05, 2010/11, and 2015/16 surveys. The HICES are repeated cross-sectional surveys that serve as the official source for poverty statistics in Ethiopia. The surveys have household consumption and expenditure information by food item and income.

Second, primary data were collected. We focus on understanding rural-urban linkages given the increasing importance of such linkages in the country. A survey of 955 dairy producers was fielded in two major dairy rural production zones around Addis Ababa, the zones of North and West Shewa; in suburban zones; and in the city of Addis Ababa in January and February 2018. 97 dairy farming households were interviewed in Addis Ababa, 256 in suburban areas in the *Oromia Special Zone surrounding Finfinne*, and 602 in rural areas. As part of the survey, we also interviewed 13 large commercial farms, defined as those with more than 25 cows. In rural areas, we ranked all woredas by remoteness to Addis Ababa. We then divided them in quartiles and selected farms randomly from each stratum proportional to the number of cows and woredas. Three kebeles (the smallest administrative division) were selected per woreda. In each selected kebele, we did a census of all households with cows in milk. We randomly selected ten households from those households that had three or more cows in milk and ten from those households that had one or two cows in milk. In each selected woreda, the plan was that 75 dairy producers were interviewed. When averages are calculated, the relative weight of each strata is taken into consideration.

After the selection of the sample households, a comprehensive survey was fielded that collected information on household characteristics, income generating activities, assets, and details on cows and dairy activities. Notably, recall questions are strongly relied upon in the analysis. The recall questions focused on main changes in the business of dairy producers and, therefore, concerned issues that likely were easy to remember. Nonetheless, we acknowledge that such recall questions are prone to measurement error [16, 17]. The recall results are therefore only used in our descriptive analyses. We do not conduct regression analysis using these recall data. To the extent possible, we also complement these recall data with other sources of information.

Table 1 provides descriptive statistics on the overall population in the study areas and on the producers that were interviewed. Note that in this table and in further tables presented in a similar manner, large farms are excluded from the aggregated household statistics for urban, suburban, and urban areas. Relying on official data, we note that 52 percent of rural farming households have cows. However, the number of cows per farm is limited. An average dairy farm in rural areas has less than 2 cows, i.e. 1.8 cows on average, and the share of dairy farms that have more than two cows is only 17 percent. The number of cows per dairy farms is slightly higher in suburban areas, at 2 cows, but far fewer suburban households own cows. In urban areas, few households own cows, but those that do have larger farms: almost 70 percent of the urban dairy farms own more than 2 cows. Surprisingly, there are a substantial number of cows in urban farms, estimated at almost 29,000. There are also about 5,000 cows on large farms of 25 cows or more, mostly located in urban and suburban areas.

Regarding the value of all cows on a farm, we see strong differences for each category driven by higher-priced cows as well as by more cows per farm in better connected areas. The average

**Table 1. Descriptive statistics of dairy farming households in study areas.**

| | | Households | | | Large farms |
|---|---|---|---|---|---|
| | Unit | Rural | Suburban | Urban | (≥ 25 cows) |
| **Overall population** | | | | | |
| Share of households with cows | % | 52 | 34 | 0.1 | |
| Of which, share of farms with more than 2 cows | % | 17 | 25 | 69 | |
| Cows per farm, average | number | 1.8 | 2.0 | 5.3 | 44.5 |
| Number of cows | total | 1,030,000 | 360,000 | 29,000 | 5,000 |
| **Sample** | | | | | |
| Sample households | number | 600 | 248 | 94 | 13 |
| Medium farms (3 cows or more) | number | 247 | 136 | 67 | |
| Small farms (1 or 2 cows) | number | 353 | 112 | 27 | |
| **Characteristics of dairy farm households,** Mean (Standard Deviation in brackets) | | | | | |
| Household size | number | 5.9(2.0) | 6.0(2.2) | 5.7(2.1) | |
| Female-headed households | % | 9.6 | 9.7 | 13.6 | |
| Age of household head | years | 48.5(15.9) | 50.6(12.5) | 53.9(13.2) | |
| Education level of household head | years | 2.8(3.5) | 3.4(3.9) | 6.4(4.0) | |
| Land area cultivated | hectares | 2.1(1.6) | 2.5(1.8) | 0.1(0.4) | |
| *Assets* | | | | | |
| Total value of cattle owned by household | Birr | 51,890 | 87,000 | 287,100 | 3,212,000 |
| | | (48,431) | (237,520) | (257,411) | (1,649,413) |
| | USD | 1,900 | 3,180 | 10,500 | 117,400 |
| | | (1,771) | (8,684) | (9,412) | (60,308) |
| Total value of cows owned by household | Birr | 18,650 | 43,000 | 205,700 | 2,193,000 |
| | | (26,146) | (215,670) | (180,825) | (1,186,011) |
| | USD | 680 | 1,570 | 7,520 | 80,200 |
| | | (956) | (7,886) | (6,612) | (43,364) |
| Total value of non-livestock assets | Birr | 3,670 | 20,200 | 216,500 | 6,747,000 |
| | | (22,147) | (116,454) | (547,616) | (17,700,000) |
| | USD | 130 | 740 | 7,920 | 246,700 |
| | | (810) | (4,258) | (20,023) | (645,714) |
| Value of non-livestock assets/value of cows | % | 20 | 47 | 105 | 308 |

Source: Authors' calculations–for urban farms in Addis Ababa, see [18]; for rural, Bureau of Livestock, Oromia

value of all cows on a farm in suburban areas (1,570 USD) is 2.5 times higher than for the rural areas (680 USD) and the value in urban areas (7,520 USD) is more than ten times higher than in rural areas. Cow herds on large farms are valued on average at more than 80,000 USD. We further see much higher non-livestock assets in suburban and urban areas than in rural ones. Moreover, if we compare the non-livestock assets to the value of cows, we see that this ratio is low in rural areas (20 percent), about equal for urban farms, but the value of non-livestock assets are over three time the value of the cow herd for large farms, indicating increasing use of non-cow capital assets for these better connected and larger dairy farms. Regarding the household characteristics, we note significantly more female-headed households managing farms in urban areas and a strong increase in educational attainment of the heads of dairy farming households in considering rural, suburban, and urban farms in sequence.

As for our methodology, we rely on a number of different models to understand associates of access to services and adoption of modern practices. We use a probit model when the dependent variable is measured as a dummy (e.g. the use of commercial pre-mixes or not).

The model is then specified as $\Pr(Y = 1|X) = \Phi(X\beta)$ where Pr denotes the probability that the dependent variable $Y = 1$, X is a vector of right-hand side variables, $\beta$ is the vector of estimated coefficients, and $\Phi$ is the Cumulative Distribution Function of the standard normal distribution. We use a tobit model when the dependent variable is measured as a proportion (e.g. share of cross-bred cows). The model can then be written as a latent regression where $Y^* = f(X)$ where $Y^*$ is not observed but a censored indicator, Z, is observed where $Z = 1$ if $Y^* \geq d$ and $Z = 0$ if $Y^* < d$. d is a censoring threshold and the observed dependent variable equals $Y^*$ if $Z = 1$. Y is missing (but converted to zero) if $Z = 0$. We also run productivity regressions of the form $Y = f(X)$ where Y is the milk yield and Y are production factors. Our preferred model are fixed-effect production functions. They have advantages over other specifications in that they control for unobserved heterogeneity at the household level. They therefore produce consistent parameter estimates. However, as a fixed effect model removes the household effect which might provide additional information, we also run pooled models.

## Results and discussion

### Downstream consumption and midstream transformation

**Dairy consumption patterns in Addis Ababa.** We rely on data from national household surveys to assess trends in consumption of dairy products in Addis Ababa. The results in Table 2 show that overall dairy consumption in Addis Ababa remains low at 10.2 kg of total

**Table 2. Consumption of dairy products in Addis Ababa, value of per adult equivalent expenditure in 2005, 2011, and 2016 by product category.**

| Year | | Unit | Cow milk | Powdered milk | Yoghurt | Cottage cheese | Butter | Total |
|------|------|------|----------|---------------|---------|----------------|--------|-------|
| **2005** | Poorest quintile | Birr | 0.3 | 0.0 | 0.0 | 0.0 | 0.2 | 0.5 |
| | Quintile 2 | Birr | 1.1 | 0.0 | 0.1 | 0.5 | 1.2 | 3.0 |
| | Quintile 3 | Birr | 2.7 | 0.1 | 0.3 | 0.7 | 5.1 | 8.9 |
| | Quintile 4 | Birr | 3.7 | 0.5 | 0.3 | 1.0 | 17.3 | 22.8 |
| | Richest quintile | Birr | 28.9 | 6.2 | 0.3 | 0.6 | 51.6 | 87.7 |
| | Total | Birr | 10.0 | 1.9 | 0.3 | 0.6 | 20.2 | 33.0 |
| | Share | % | 30.3 | 5.8 | 0.8 | 1.9 | 61.2 | 100.0 |
| | *Quantities* | *kg* | *6.0* | *0.1* | *0.1* | *0.3* | *1.4* | *7.8* |
| **2011** | Poorest quintile | Birr | 0.0 | 0.0 | – | – | 0.2 | 0.2 |
| | Quintile 2 | Birr | 0.4 | 0.1 | 0.1 | 0.1 | 3.0 | 3.5 |
| | Quintile 3 | Birr | 2.0 | 0.3 | 0.3 | 0.4 | 7.4 | 10.4 |
| | Quintile 4 | Birr | 12.1 | 0.8 | 0.6 | 1.5 | 21.9 | 36.9 |
| | Richest quintile | Birr | 69.8 | 14.6 | 2.6 | 8.2 | 68.6 | 163.8 |
| | Total | Birr | 22.1 | 4.2 | 0.9 | 2.6 | 25.6 | 55.4 |
| | Share | % | 39.9 | 7.6 | 1.7 | 4.7 | 46.1 | 100.0 |
| | *Quantities* | *kg* | *7.7* | *0.4* | *0.3* | *0.8* | *1.2* | *10.4* |
| **2016** | Poorest quintile | Birr | 2.5 | 1.8 | 0.2 | 0.4 | 23.9 | 28.8 |
| | Quintile 2 | Birr | 16.5 | 0.7 | 1.6 | 1.1 | 24.3 | 44.3 |
| | Quintile 3 | Birr | 20.7 | 2.9 | 1.7 | 3.1 | 31.5 | 59.9 |
| | Quintile 4 | Birr | 47.1 | 7.7 | 0.7 | 4.4 | 27.5 | 87.4 |
| | Richest quintile | Birr | 69.5 | 17.8 | 3.1 | 17.9 | 40.1 | 148.3 |
| | Total | Birr | 37.5 | 7.6 | 1.7 | 6.9 | 30.8 | 84.5 |
| | Share | % | 44.4 | 9.0 | 2.0 | 8.2 | 36.5 | 100.0 |
| | *Quantities* | *kg* | *8.5* | *0.1* | *0.3* | *0.5* | *0.8* | *10.2* |

Birr for the three years are expressed in real 2011 Birr

Source: Authors' calculations from HCES (2005, 2011, 2016)

dairy products and 8.5 liters of liquid milk per adult equivalent annually (Average annual global milk consumption is estimated at 111 liters per capita [19]). However, the annual quantities consumed per adult equivalent increased by 31 percent between 2005 and 2016. Some notable shifts within the consumption of dairy products are noted as well. Liquid milk consumption increased significantly. While the share of cow milk in dairy expenditures was 30 percent in 2005, this increased to 44 percent in 2016. On the other hand, butter consumption is on the decline, decreasing from 61 percent of the dairy budget in 2005 to 36 percent in 2016. We also see an increasing importance of imported powder milk, as its share in the dairy budget increased from 6 percent in 2005 to 9 percent in 2016. At the country level, the value of powdered milk imports increased rapidly, amounting to almost 20 million USD in 2015 from just over 5 million USD in 2005.

We further note significant differences in dairy consumption between poor and rich households. Dairy consumption expenditures in 2016 were five times higher for the richest quintile compared to the poorest one. We see this strong gradient over welfare quintiles for almost all dairy products. The gradient is the least for butter. The growth in income in the country and in Addis Ababa–poverty headcount levels in Addis Ababa were 17 percent of the population in 2016 compared to 32 percent in 2005 [20]–partly explains the higher consumption of such dairy products, given positive income elasticities for ASFs [21]. This is linked to Bennet's law that states that as income increases, not only does the quantity of food increase less than proportionally but also the composition of the food basket changes. Specifically, the consumption of starchy staple food declines and that of high-value foods such as ASF increases.

On top of consumption growth per capita, the Addis Ababa metropolis is also rapidly growing in population, leading to increasing demand. Based on an agglomeration method, [22] evaluated the population of the Addis Ababa metropolitan area at 4.12 million in 2015. Using historical data, they estimated the population to have been 3.42 million in 2007, or an annual increase of the metropolitan area population by 2.4 percent.

The upshot is that a significant increase in dairy consumption has been noted in the metropolitan area of Addis Ababa in the last decade because of important income gains as well as population growth.

**The dairy processing sector.** With growth in dairy consumption, we also see increasing formalization of dairy markets. At the national level, there were eight dairy processing companies active in 2007. By 2017, this number had more than tripled to 25 (Fig 1). Thus, there were large investments in the last decade, and more are planned. We obtained data on processing and the processing capacity of the dairy processing companies in Addis Ababa and surrounding areas. In the period 2016/17, daily processing was almost 200,000 liters of milk per day. There is significant concentration, with the four largest dairy processing firms supplying three-quarters of all the pasteurized dairy products in the market. We also find significant overcapacity in the sector, with over 40 percent of the dairy processing capacity not being used. It is to be noted that, in contrast to a number of other countries, such as India, processing by dairy cooperatives is relatively less important. The largest cooperative active in the areas around Addis Ababa is the Ada'a cooperative with a market share of about 5 percent.

## Upstream production of dairy products

**Structure of dairy production.** Table 3 shows estimates of the share in the liquid milk supply (overall and pasteurized) for households by location and for large farms, which are defined as those that have 25 cows or more. Several insights are obtained. First, smallholder dairy farms supply most of the liquid milk to the city. Their share is evaluated at 89 percent of all the milk supplied to the city. However, we see increasing upscaling, especially so in

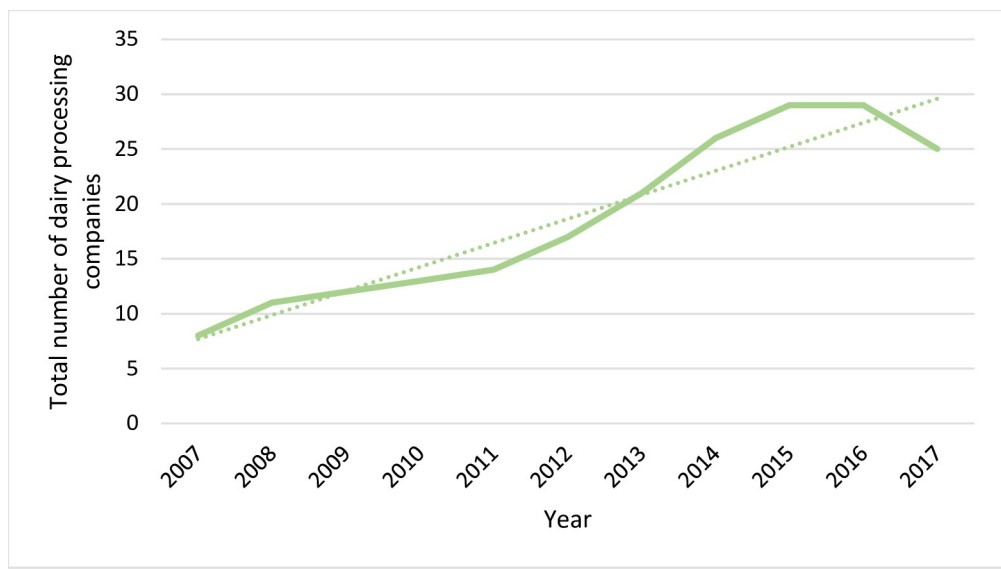

**Fig 1. Number of processing firms nationally.** Source: Ethiopian Meat and Dairy Industry Development Institute.

suburban areas. While there were on average 0.67 farms with more than 25 cows per woreda in 2007, this number had increased eight-fold in 2017. Overall, it is estimated that in the year prior to the survey, 128 large farms were active in Addis Ababa and suburban areas. We estimate that they make up 11 [16] percent of the liquid (pasteurized) milk supply to the city (Table 3). While these farms are relatively minor in the overall milk value chain to the city, they make up already a sizable and dynamic portion of the value chain supplying the capital. Similar processes of larger farms supplying cities have been documented elsewhere [15, 23, 24].

Second, urban dairy farms are surprisingly important. As reported earlier, it is estimated that there are almost 29,000 dairy cows in the city supplying 31 percent of all the urban liquid milk (including the commercial farms in the city). The suburban areas supply 26 percent of all the liquid milk. Rural areas make up 37 and 44 percent of the overall and pasteurized liquid milk market, respectively.

Third, we see significant differences in milk yields, measured in liter produced per cow per day, by type of farm and location. Large farms obtain yields that are significantly higher than those of smallholders. Mean yields for large farms were 18.6 l/cow/day. This compares to 3.3 to 4.8 l/day in rural and suburban areas respectively. Urban dairy farming households also are able to achieve much higher yields than rural and suburban households.

Fourth, we collected detailed labor data by gender, by age, and by family and hired labor on how much time was spent daily on milking cows in the morning and the evening, marketing milk, preparing feeds and feedings cows, cleaning the sheds and utensils, and on taking cows out for watering and grazing. We find that labor productivity is significantly higher for large farms and for those farmers that are better connected to markets. Productivity per hour worked more than doubles for suburban dairy farming households compared to rural ones. Large farms have labor productivity that is more than fifteen times higher than suburban farmers. Such efficiency gains for the bigger farms have been shown before in these settings and have led some to argue for a more intensive focus on stimulating the emergence of such farms in the African context [25].

Fifth, we see important dynamics over time. We rely on administrative and recall questions from households and community focus groups. Administrative data show that milk yields are

**Table 3. Liquid milk supply to Addis and yields.**

| | Unit | Time period | Households | | | Large farms (≥ 25 cows) |
|---|---|---|---|---|---|---|
| | | | Rural | Suburban | Urban | |
| *Administrative data* | | | | | | |
| Farms with more than 25 cows, average per woreda | number | 2017 | 0.13 | 5.33 | | |
| | | 2007 | 0.13 | 0.67 | | |
| Large farms | number | | 16 | 62* | 50** | 128 |
| Liquid milk, share of total | share (%) | time of survey | 37 | 26 | 26 | 11 |
| Pasteurized milk, share of total | share (%) | time of survey | 44 | 28 | 11 | 16 |
| *Milk yields* | | | | | | |
| *Administrative data* | | | | | | |
| Mean yields: Cross-bred cows | l/day/cow | time of survey | 11.0 | 10.4 | | |
| | | 10 years earlier | 7.5 | 10.2 | | |
| Mean yields: Local cows | l/day/cow | time of survey | 2.9 | 1.9 | | |
| | | 10 years earlier | 2.5 | 1.4 | | |
| *Survey farm households* | | | | | | |
| Mean milk yields during lactation | l/day/cow | time of survey | 3.3 | 4.8 | 9.9 | 18.6 |
| | | 10 years earlier | 2.6 | 4.0 | 9.6 | 15.8 |
| Median milk yields during lactation | l/day/cow | time of survey | 1.7 | 2.2 | 9.7 | 18.3 |
| | | 10 years earlier | 1.5 | 2.0 | 8.7 | 14.0 |
| Mean dry months per cow | number | time of survey | 7.7 | 7.1 | 3.5 | 3.2 |
| Median dry months per cow | number | time of survey | 7.0 | 6.0 | 2.0 | 3.0 |
| *Survey focus groups* | | | | | | |
| Mean milk yields during lactation | l/day/cow | time of survey | 2.5 | 3.0 | | |
| | | 10 years earlier | 1.9 | 2.8 | | |
| Median milk yields during lactation | l/day/cow | time of survey | 1.8 | 2.4 | | |
| | | 10 years earlier | 1.7 | 1.7 | | |
| *Labor productivity (gross)* | | | | | | |
| Mean milk per labor hour | l/hour | time of survey | 0.94 | 2.01 | 10.23 | 35.81 |

Based on data from urban and suburban dairy farms, excluding large farms (25 cows or more). Urban dairy farms including large farms supply 31 percent of all liquid milk to Addis Ababa.

* Including Holeta and Debre Zeyt [18]

** 13 percent of cows are found in farms of 25 cows or more

Source: Authors' calculations

significantly higher for cross-bred versus local cows and that they have generally improved over time for all categories. However, yields of local cows in rural areas have shown little growth, with increases registered of only 0.4 liter/cow/day over the previous ten years. The recall questions for households and community groups show similar patterns. We see important improvements in yields, but starting from a low base, especially for the suburban and rural households.

Relying on the household survey data, Figs 2 and 3 illustrate how annual milk yields per dairy cow vary over space and over time and by dairy herd size. To calculate productivity, we converted the milk yields reported by the households for each quarter of the lactation period and then included the average number of dry months observed for cows at the time of the survey for each household. Table 3 shows that the number of dry months is substantially higher for rural farms compared to urban and large farms.

We see an important decline of milk yields by travel cost to Addis Ababa. Yields are five times higher for dairy farms close to the city compared to those far out. Moreover, yields are

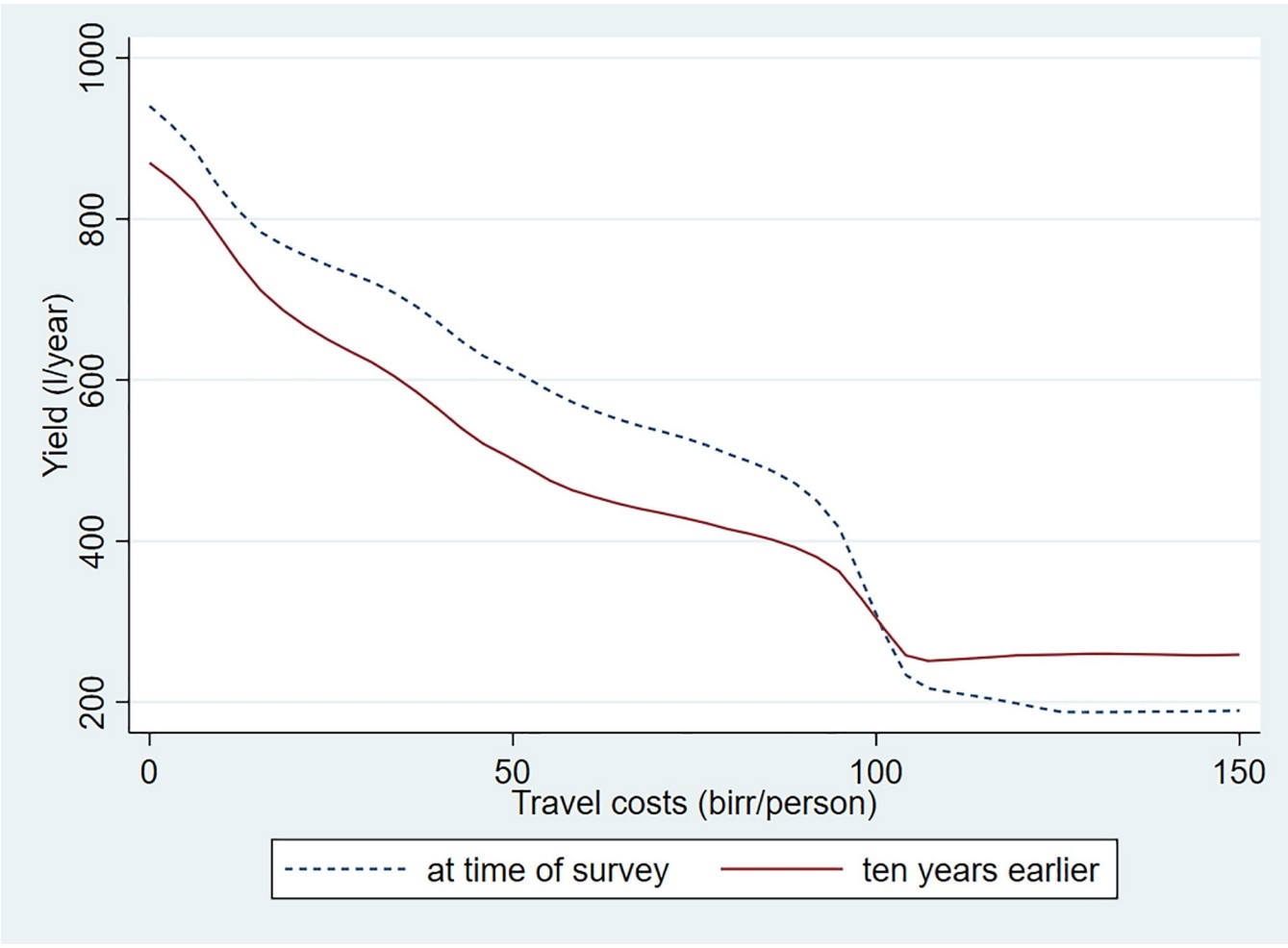

**Fig 2. Milk cow yields by travel cost at time of survey and ten years earlier.** Source:Authors' calculations.

going up over time in well-connected areas, while they are declining in the more remote areas. We further note that bigger farms have significantly higher yields and that they have shown significant improvements over time. On the other hand, small farms with only one or two cows did not show any improvements in milk yields. We discuss below reasons for this important observed dynamic. We will focus on input markets and look at the adoption of cross-bred cows, feeding practices, and access to livestock services.

**Dairy inputs and services.** Improvement of breeds is an important way to raise the genetic potential and productivity of cows. Better performing dairy breeds–such as Holstein and Jersey–are cross-bred with local cows to achieve higher milk yields. This is increasingly done through artificial insemination (A.I.) which has shown major positive results for the spread of cross-bred cows in Ethiopia and elsewhere [26]. A.I. must be done by specialized technicians and a supply chain needs to be set up that delivers semen in cold conditions at the farm. The government has invested significantly in recent years to improve A.I. distribution systems and uptake in the country. Data were obtained from the Ministry of Agriculture and Livestock on levels and changes in A.I. uptake in the country. Table 4 shows the evolution in the production, distribution, insemination, pregnancy, and calves born using A.I. for the country as whole as well as for Oromia region, where the surveyed rural and suburban areas are

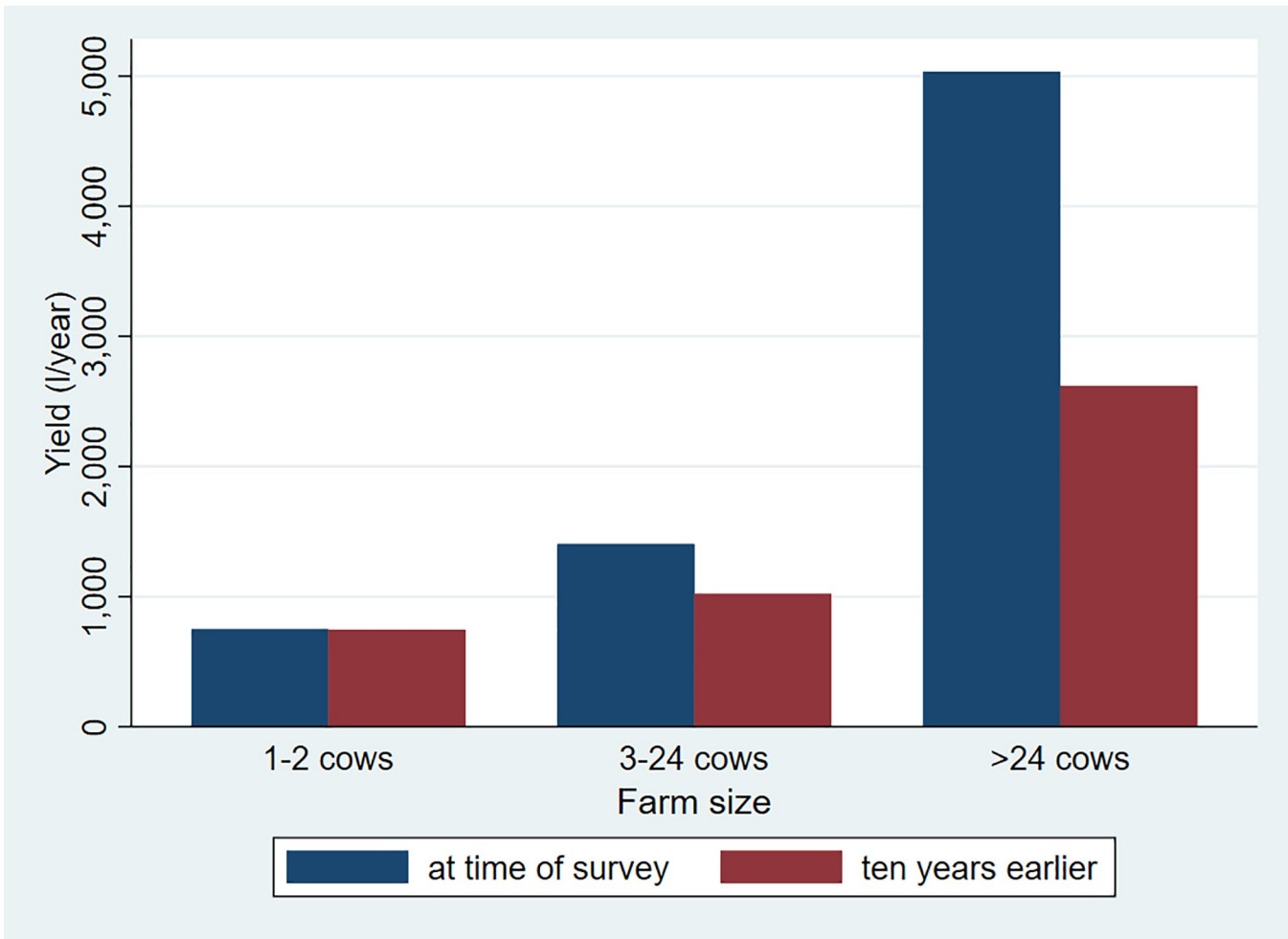

**Fig 3. Milk cow yields by dairy herd size at time of survey and ten years earlier.** Source:Authors' calculations.

located. The number of calves born using A.I. was 10 times higher in 2016 compared to 2007. Despite the rapid increase, only 230,000 calves were born on average annually through A.I. between 2015 and 2017, still a small number compared to the total cattle population in Ethiopia. More than one quarter of the A.I. semen was distributed in Oromia region.

In the regions where we fielded the survey, data on the number of A.I. agents present in the woreda were collected from administrative sources. The number of A.I. agents available in each woreda almost tripled between 2008 and 2017 (Table 5). To assess the effects on the

**Table 4. Artificial insemination production and distribution in Ethiopia and Oromia.**

|  | Year | Production | Distribution | Insemination | Calves born |
|---|---|---|---|---|---|
| Artificial insemination at national level | 2007 | 118,948 | 124,246 | 71,103 | 24,181 |
| 3-year moving average | 2016 | 842,185 | 825,335 | 386,049 | 229,618 |
| Artificial insemination in Oromia region | 2012 |  | 13,912 |  |  |
|  | 2017 |  | 232,144 |  |  |

Source: Ministry of Agriculture and Livestock

farmer, questions were asked to the households on the availability of A.I. agents and A.I. semen. We see that significant change occurred over the period studied. For example, while 51 percent of the suburban dairy farming households reported that A.I. agents were not available for them ten years before the survey, this figure decreased to 24 percent at the time of the survey. We also see much greater availability in rural villages as well. However, almost 40 percent of rural dairy farming households still reported not to have access to A.I. agents at the time of the survey. The availability of A.I. semen shows similar patterns.

While access to A.I. has improved, the share of farmers that use A.I. and that have cross-bred cows is still limited, especially so in remote rural areas. 27 percent of the rural dairy farmers had cross-bred cows at the time of survey, compared to 98 percent in urban areas and 100 percent for commercial farms (Table 5). This difference is seemingly driven by differential access to A.I., but also by the high costs of acquisition, the maintenance of cross-bred cows (the feeding costs), as well as the higher risks for holding such cows, as they are generally perceived to be more vulnerable than local cows to disease and climatic shocks. The farms that adopt such cross-bred cows are therefore ones that have generally higher risk-bearing capacity. Overall, we see important changes in the adoption of cross-bred cows over the past ten years with an increase of 7 and 11 percentage points in suburban and rural areas, respectively.

A second factor possibly explaining higher milk yields is improvements in feed for dairy cows. We see important changes in the feed sector, seemingly driven by both a desire to supply more milk to emerging urban dairy markets and increasing constraints in grazing land

**Table 5. Cross-bred cows and artificial insemination access.**

| Variable | Unit | Time period | Households | | | Large farms (≥ 25 cows) |
|---|---|---|---|---|---|---|
| | | | Rural | Suburban | Urban | |
| *Household survey* | | | | | | |
| Farms that have cross- bred cows | % | time of survey | 27.0 | 26.6 | 97.9 | 100.0 |
| | | 10 years earlier | 15.8 | 19.5 | 96.1 | 100.0 |
| *Availability of A.I. agent* | | | | | | |
| Mostly available | % | time of survey | 31.0 | 30.4 | 68.5 | 84.6 |
| | | 10 years earlier | 4.6 | 9.1 | 42.9 | 46.2 |
| Sometimes available | % | time of survey | 29.8 | 45.8 | 30.1 | 15.4 |
| | | 10 years earlier | 28.7 | 40.2 | 41.0 | 46.2 |
| Not available | % | time of survey | 39.2 | 23.8 | 1.4 | 0.0 |
| | | 10 years earlier | 66.7 | 50.7 | 16.1 | 7.7 |
| *Availability of cross-bred semen* | | | | | | |
| Mostly available | % | time of survey | 31.6 | 34.4 | 50.2 | 63.6 |
| | | 10 years earlier | 5.7 | 10.6 | 32.7 | 45.5 |
| Sometimes available | % | time of survey | 32.4 | 43.2 | 45.2 | 27.3 |
| | | 10 years earlier | 27.1 | 45.2 | 52.7 | 54.6 |
| Not available | % | time of survey | 35.9 | 22.5 | 4.6 | 9.1 |
| | | 10 years earlier | 67.2 | 44.2 | 14.7 | 0.0 |
| Median travel time to access an A.I. agent | minutes | time of survey | 90.0 | 60.0 | 20.0 | 10.0 |
| | | 10 years earlier | 90.0 | 60.0 | 20.0 | 20.0 |
| *Administrative data* | | | | | | |
| Mean number of A.I. agents in woreda | number | 2008 | 0.71 | 1.00 | | |
| | | 2017 | 1.91 | 2.67 | | |

A.I. = artificial insemination.

Source: Authors' calculations

**Table 6. Dairy feed sector.**

| | Unit | Time period | Households | | | Large farms (≥ 25 cows) |
|---|---|---|---|---|---|---|
| | | | Rural | Suburban | Urban | |
| *Most common feeding system (dry season)* | | | | | | |
| Only rely on grazing (free-range/tethered) for feed | % | time of survey | 31.8 | 30.0 | 2.8 | 0.0 |
| | | 10 years earlier | 53.2 | 39.5 | 7.0 | 7.7 |
| Only rely on stall feeding for feed | % | time of survey | 11.1 | 13.8 | 86.5 | 92.3 |
| | | 10 years earlier | 8.3 | 9.4 | 79.5 | 69.2 |
| Mainly grazing with some stall feeding | % | time of survey | 38.2 | 43.1 | 3.3 | 7.7 |
| | | 10 years earlier | 29.5 | 38.7 | 5.3 | 15.4 |
| Mainly stall feeding with some grazing | % | time of survey | 18.9 | 13.1 | 7.4 | 0.0 |
| | | 10 years earlier | 9.0 | 12.5 | 8.1 | 7.7 |
| *Access to grazing land* | | | | | | |
| Communal grazing land not available | % | time of survey | 42.5 | 57.0 | 60.7 | 30.8 |
| | | 10 years earlier | 15.7 | 31.6 | 45.1 | 30.8 |
| Private grazing land not available | % | time of survey | 21.7 | 12.4 | 64.3 | 25.0 |
| | | 10 years earlier | 9.1 | 7.4 | 46.9 | 25.0 |
| *Type of feed farmers mostly purchase* | | | | | | |
| Wheat bran | % | time of survey | 16.1 | 38.2 | 91.4 | 69.2 |
| | | 10 years earlier | 8.4 | 25.4 | 87.2 | 76.9 |
| Oilseed cake | % | time of survey | 14.9 | 20.1 | 43.3 | 53.9 |
| | | 10 years earlier | 6.7 | 15.5 | 60.8 | 69.2 |
| Molasses | % | time of survey | 0.5 | 5.3 | 18.7 | 46.2 |
| | | 10 years earlier | 0.6 | 4.3 | 35.7 | 46.2 |
| Commercial pre-mix | % | time of survey | 0.8 | 5.8 | 41.0 | 53.9 |
| | | 10 years earlier | 0.4 | 5.4 | 37.6 | 69.2 |

Source: Authors' calculations

(Table 6). In rural areas, 53 percent of the dairy households relied only on grazing area ten years before the survey, but this declined to 32 percent at the time of the survey. Increasing lack of both private and communal grazing areas seemingly explains this trend. While 16 percent of rural dairy farming households said that communal grazing land was not available ten years before the survey, that increased to 42 percent at the time of the survey. Stall feeding is becoming relatively more common in all areas, but exclusive stall feeding is overall still relatively rare, except in urban areas and for commercial farms.

To assure the increasingly required stall feeding, farmers could use own or purchased feed. The data show that commercial feed markets are rapidly taking off, especially so in better connected areas. In these areas, wheat bran and oilseed cakes have quickly been adopted as the preferred feeds for stall feeding of dairy cows. Thirty-eight and 20 percent of the farmers stated that they commonly purchase these products in suburban areas, respectively. Commercial pre-mixed brands are used, but only a small proportion of the dairy farmers have adopted them, i.e., 1, 6, and 41 percent for rural, suburban, and urban areas, respectively. Commercial pre-mixed feed is mostly used by commercial farms– 54 percent of them use such feed. This increasing demand for commercial feed has led to the emergence of a commercial feed selling industry and more choice in commercial feed sellers for farmers.

Finally, over the last decade, there have been significant changes in the set-up of livestock extension and health animal service delivery in the country. Based on administrative data, Table 7 shows how the number of extension agents and of people employed in clinics and

**Table 7. Access to health and extension services.**

| | Unit | Time period | Households | | | Large farms (≥ 25 cows) |
|---|---|---|---|---|---|---|
| | | | Rural | Suburban | Urban | |
| *Administrative data* | | | | | | |
| Livestock agents per woreda, average | number | time of survey | 18.8 | 26.5 | | |
| | | 10 years earlier | 23.4 | 11.0 | | |
| Animal health experts per woreda, average | number | time of survey | 17.9 | 15.7 | | |
| | | 10 years earlier | 4.0 | 8.0 | | |
| Functioning animal health clinics per woreda, average | number | time of survey | 9.3 | 12.0 | | |
| | | 10 years earlier | 3.9 | 7.0 | | |
| Pharmacies per woreda, average | number | time of survey | 4.1 | 7.0 | | |
| | | 10 years earlier | 0.8 | 2.3 | | |
| Vaccination rate cattle* | number | time of survey | 0.74 | 0.54 | | |
| | | 10 years earlier | 0.09 | 0.11 | | |
| *Household survey* | | | | | | |
| *Availability of extension agents* | | | | | | |
| Not available | % | time of survey | 19.8 | 9.4 | 9.5 | 0.0 |
| | | 10 years earlier | 48.9 | 30.1 | 19.9 | 0.0 |
| Sometimes available | % | time of survey | 43.6 | 45.7 | 43.5 | 25.0 |
| | | 10 years earlier | 46.1 | 57.9 | 52.4 | 41.7 |
| Mostly available | % | time of survey | 36.5 | 44.9 | 47.0 | 75.0 |
| | | 10 years earlier | 5.1 | 12.0 | 27.7 | 58.3 |
| *Availability of veterinary and animal health worker* | | | | | | |
| Not available | % | time of survey | 11.7 | 6.2 | 1.7 | 8.3 |
| | | 10 years earlier | 37.9 | 26.2 | 4.6 | 8.3 |
| Sometimes available | % | time of survey | 47.1 | 43.2 | 44.8 | 8.3 |
| | | 10 years earlier | 54.4 | 60.9 | 60.4 | 66.7 |
| Mostly available | % | time of survey | 41.1 | 50.6 | 53.5 | 83.3 |
| | | 10 years earlier | 7.7 | 12.9 | 35.0 | 25.0 |
| *Availability of medicines for dairy animals* | | | | | | |
| Medicines at nearest pharmacy not available | % | time of survey | 2.1 | 1.6 | 1.5 | 0.0 |
| | | 10 years earlier | 18.5 | 14.1 | 1.5 | 0.0 |
| Median travel time to nearest animal health post | minutes | time of survey | 60.0 | 60.0 | 30.0 | 5.0 |
| | | 10 years earlier | 90.0 | 60.0 | 30.0 | 5.0 |
| Median travel time to get animal medicines | minutes | time of survey | 60.0 | 60.0 | 30.0 | 35.0 |
| | | 10 years earlier | 90.0 | 60.0 | 30.0 | 50.0 |

* Average share of anthrax, black leg, bovine pasteurolis and lumpy skin disease vaccination rates.

Source: Authors' calculations

pharmacies increased over time. This has led to a significant reduction in the travel time farmers incur to get to health posts and clinics and to pharmacies, improved availability of veterinarians and animal health extension workers, and improved availability of medicines. All these indicators have improved over the last decade (Table 7). Because of better access to health services for their dairy cows, different measures on health indicators have also shown progress. First, vaccination rates have increased significantly over the last decade. Second, [27] show, based on national data from CSA, that livestock death rates declined over the last decade. They found that relative to 2005, death rates in 2014 were 2 percent lower for cattle. However, they also show that the number of livestock lost to deaths is more than twice the

number sold for meat production, indicating that animal health service provision is still an important challenge for the sector.

## Dairy technology adoption and productivity analysis

**Graphical analysis.** We note strong heterogeneity in patterns of adoption of different inputs in dairy production. To illustrate heterogeneity by remoteness from Addis Ababa (measured in travel costs), we rely on non-parametric regressions. These regressions do not require specifying a functional form of a relationship in advance and they help explore relationships in data without preconditions. More in particular, we implement a locally weighted scatterplot smoothing technique. We start with an analysis of access to services. We first present a graph where use of extension services is plotted over time and space and by farm size (Figs 4 and 5). Both graphs show a strong improvement in access to dairy-related extension services over time. However, we note that farmers at a greater travel cost from Addis Ababa have significantly less access to extension services than those located close to the city. While almost all dairy farmers in well-connected areas had access to livestock extension, only about 30 percent of farmers in the most remote areas reported availability of livestock extension services.

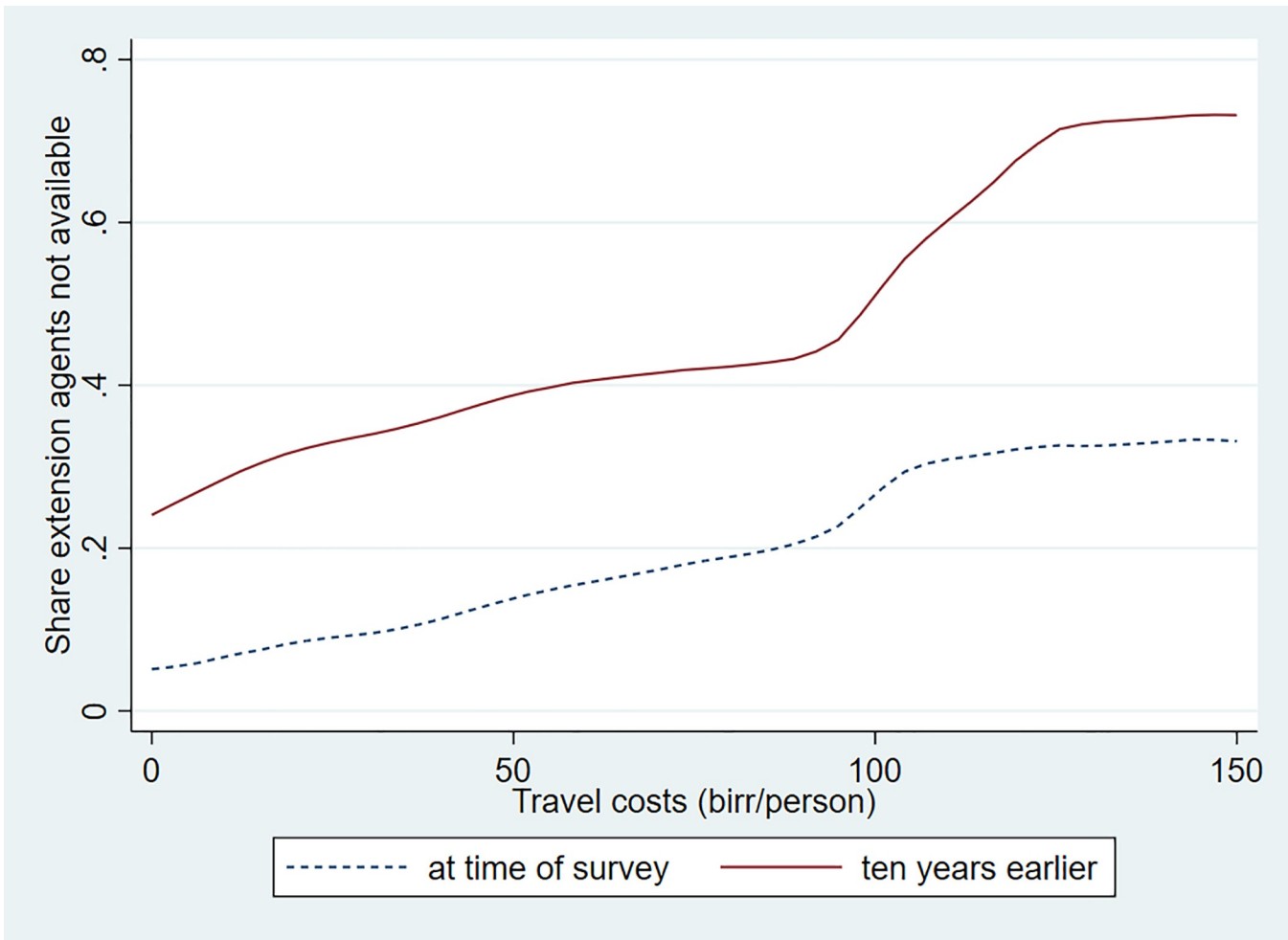

**Fig 4. Availability of livestock extension services by travel cost at time of survey and ten years earlier.** Source: Authors' calculations.

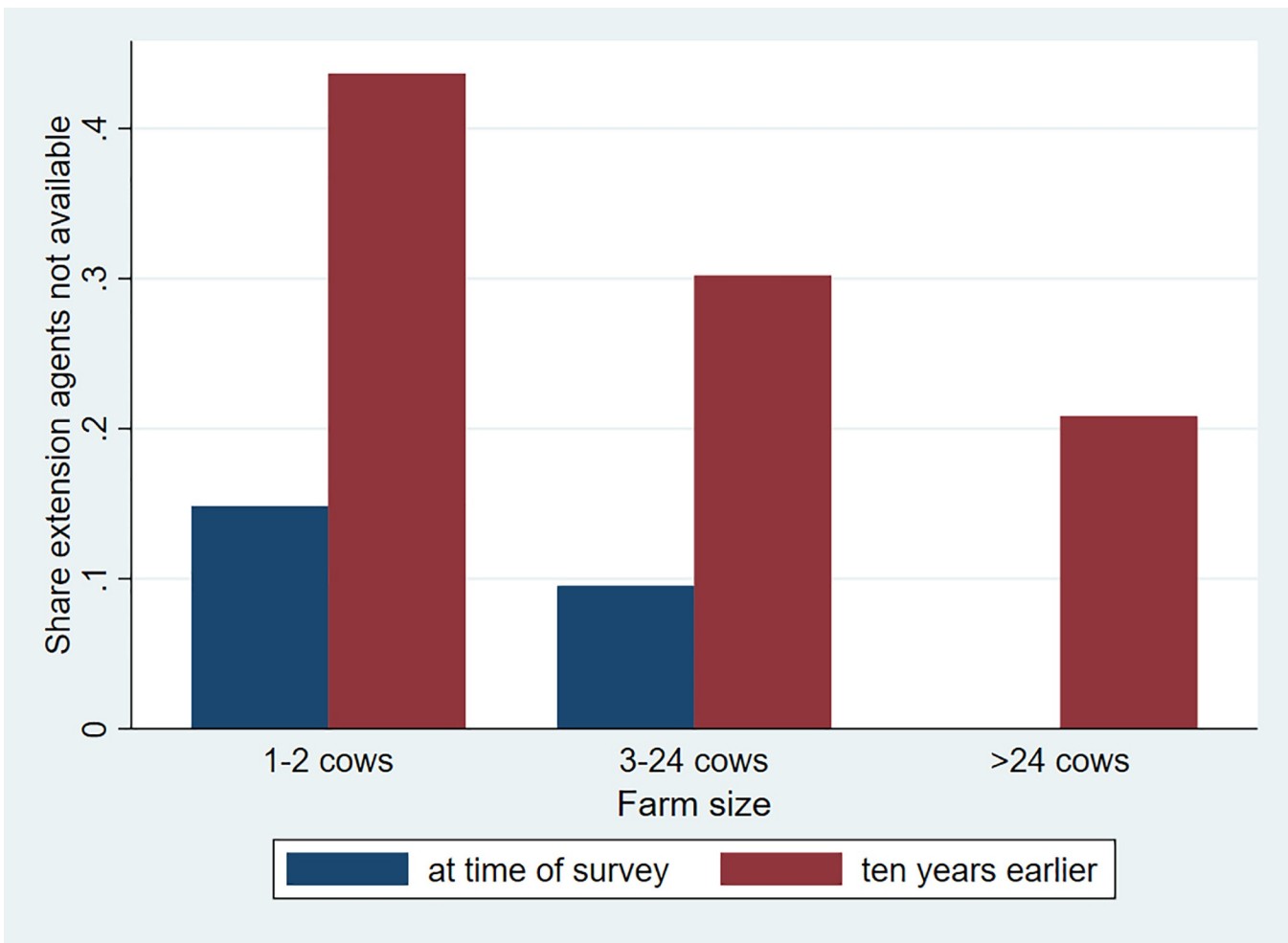

**Fig 5. Availability of livestock extension services by dairy herd size at time of survey and ten years earlier.** Source: Authors' calculations.

Use of extension services also shows a strong gradient by size of the farm with larger farms much better covered. This gradient had become smaller over the ten years prior to the survey and extension services were more equitably delivered than a decade earlier. However, still more than 15 percent of the smallest dairy farmers reported to not have received extension services at the time of the survey. These numbers are significantly smaller for the medium and large farms.

Fig 6 shows how the level and changes in the adoption rate of cross-bred cows vary over the travel cost to Addis Ababa. First, we see a clear spatial gradient. Cross-bred dairy cows are much more common in areas close to the city. Second, we see important changes in adoption of cross-bred cows over time. Moreover, we see strong patterns in the adoption of cross-bred cows by dairy herd size (Fig 7). While the share of cross-bred cows for small farms is around 30 percent, this increases significantly by herd size. The share of cross-bred cows is almost 50 percent for herd sizes of 3 to 24 cows and large dairy farm herds are almost exclusively made up of cross-bred cows. We further see an overall improvement in the adoption of cross-bred cows for medium as well as large farms. However, such improvements are relatively minor for the small farms with only one or two cows.

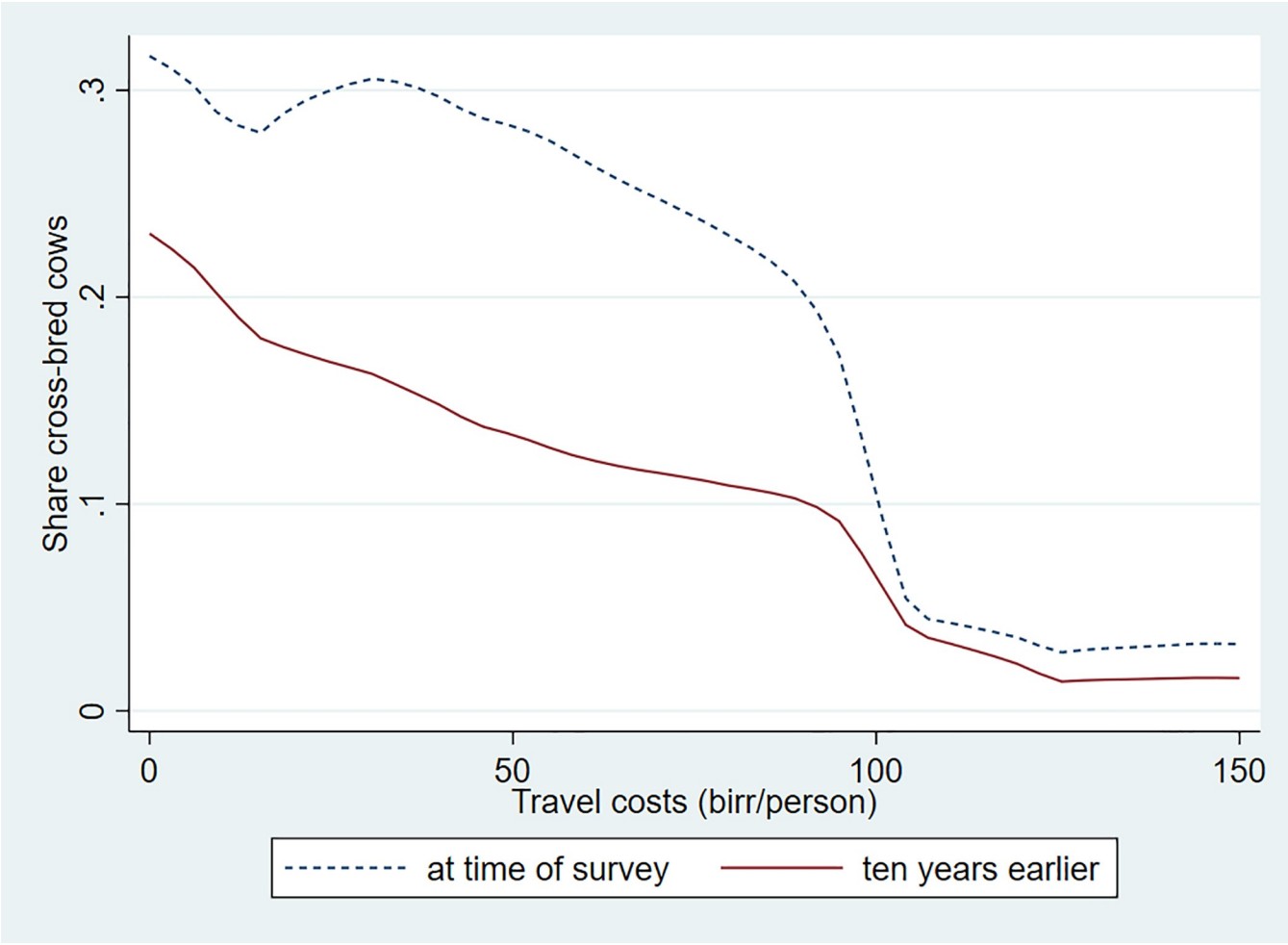

**Fig 6. Adoption of cross-bred cows by travel cost at time of survey and ten years earlier.** Source: Authors' calculations.

**Associates of adoption or access to improved dairy inputs and services.**   Given the importance of adopting new technologies in order to improve dairy productivity and of providing services that might stimulate this, we are interested to understand factors associated with adoption of improved technologies in milk production as well as access to dairy-related extension services. While non-parametric models give insights on the importance of some factors, analysis of adoption decisions is better addressed in a multi-variable regression framework to allow us to control for multiple variables at once. We therefore model access to services and the adoption decision as a probit model or, when intensity of adoption is measured, as a Tobit model to estimate what factors are associated with each adoption category in contrast to the benchmark category of no adoption. We focus especially on inclusiveness of smallholders and remote farms.

In Tables 8 and 9, we present the results of these regressions. We use as the dependent variable for our probit regressions a dummy variable that equals one in the case of uptake or use at the time of the survey and zero otherwise, while for Tobit regressions we use the share of cross-bred cows over all cows. We put on the right side of the regression several variables, including household characteristics, remoteness, and herd size, that may be associated with the use of services and adoption of these improved practices. We present for each dependent

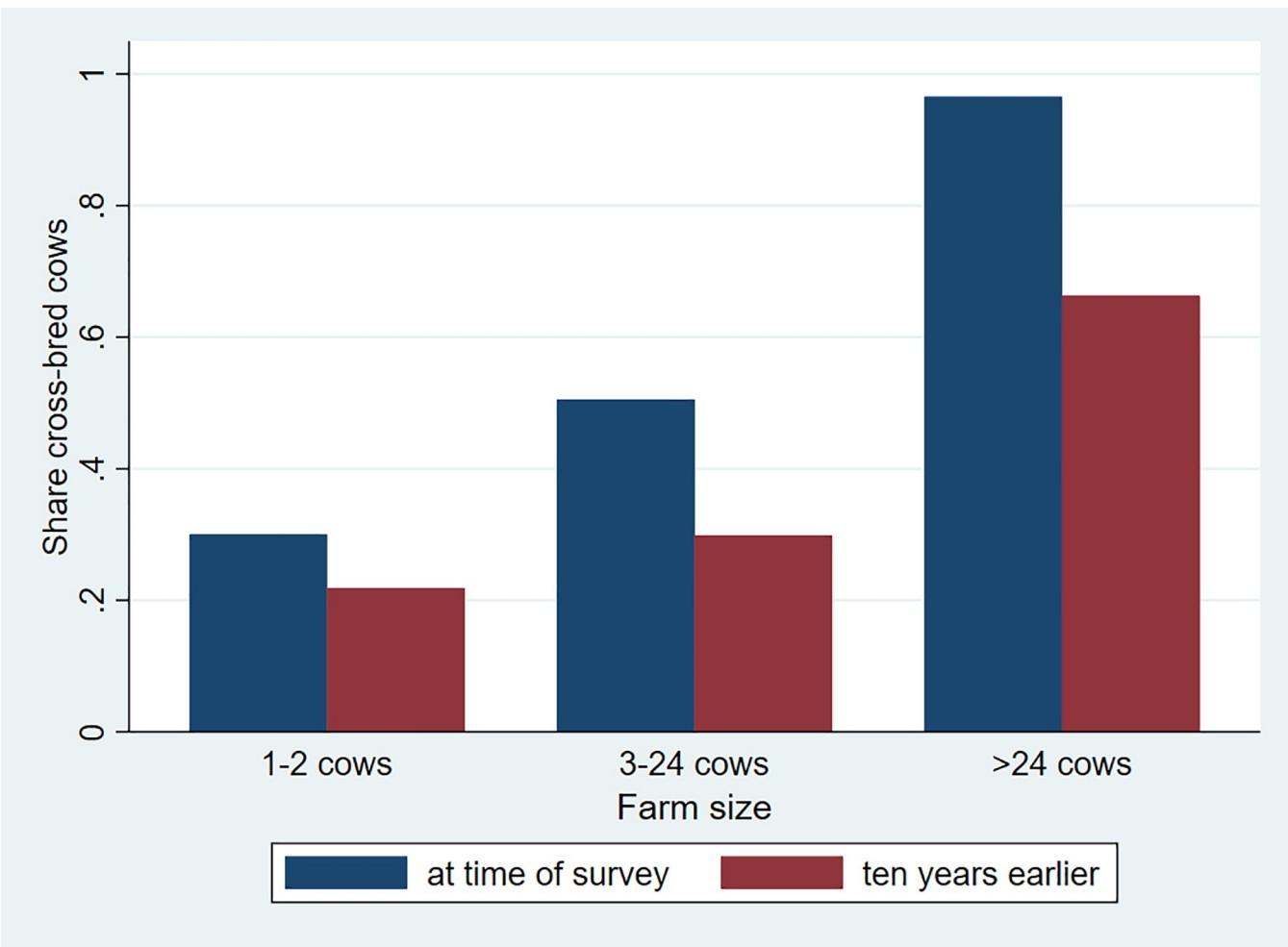

**Fig 7. Adoption of cross-bred cows by dairy herd size at time of survey and ten years earlier.** Source: Authors' calculations.

variable two specifications: one with a lower number of observations where we include household characteristics as independent variables and one with the full set of data where we do not include household characteristics, as these characteristics were not collected for the large farms. Several patterns emerge from the results of these regressions.

First, proximity to the city is a significant associate of adoption of cross-bred cows and use of purchased wheat bran for feed and commercial pre-mixes (Table 9). The magnitudes of the estimated coefficients are all sizable. A 10 percent increase of the travel cost to the city reduces the share of cross-bred cows by 11.5 percent, when commercial farms are not included, and 11.8 percent when they are (Table 9). It also reduces the probability of using purchased wheat bran by 2.3 to 2.4 percent and of using commercial pre-mixes by 0.7 to 1 percent, depending on the specification (Table 9). A 10 percent increase of transportation costs is further associated with a 1 and 0.4 percent lower probability in the use of extension agents and animal health workers respectively (Table 8). It also increases the travel cost that farmers face to obtain medicines by about 4 percent.

Second, farm size, as measured by the number of cows on the farm, also shows a strong association with most of these variables. To assess its impact, we use a dummy for farms that have 1 or 2 cows, medium-sized farms that have 3 to 24 cows, and large farms with 25 cows or

**Table 8. Factors associated with access to dairy related services.**

| Variables | Unit | Mean | Standard deviation | Use of extension agent: Probit (Average marginal effect) | | | | Access animal health worker: Probit (Average marginal effect) | | | | Walking time medicines: Log (minutes)—OLS | | | |
|---|---|---|---|---|---|---|---|---|---|---|---|---|---|---|---|
| | | | | (1) | | (2) | | (3) | | (4) | | (5) | | (6) | |
| | | | | Coefficient | t-value | Coefficient | t-value | Coefficient | t-value | Coefficient | t-value | Coefficient | t-value | Coefficient | t-value |
| Dependent variables | mean | | | 0.46 | | | | 0.89 | | | | 3.77 | | | |
| | (std. dev.) | | | | | | | | | | | (1.02) | | | |
| *Independent variables* | | | | | | | | | | | | | | | |
| Transport cost to Addis | ln(Birr +1) | 3.40 | (1.34) | -0.10** | -2.36 | -0.10*** | -2.62 | -0.04* | -1.93 | -0.04* | -1.96 | 0.38*** | 2.96 | 0.42*** | 2.94 |
| Farm size (Default: small farm) | | | | | | | | | | | | | | | |
| Medium farm (3–24 cows) | yes = 1 | 0.47 | | 0.05 | 1.33 | 0.09** | 2.53 | 0.09*** | 3.19 | 0.11*** | 4.12 | -0.13 | -1.21 | -0.15 | -1.52 |
| Large farm (≥25 cows) | yes = 1 | 0.01 | | | | 0.50*** | 2.93 | | | -0.02 | -0.31 | | | -0.56 | -1.18 |
| Gender of household head | male = 1 | 0.92 | | 0.03 | 0.42 | | | -0.03 | -0.64 | | | 0.20* | 1.80 | | |
| Age of household head | number | 48.93 | (13.70) | 0.00 | 0.02 | | | 0.00*** | 3.01 | | | -0.00 | -1.38 | | |
| Education of household head | number | 3.68 | (4.27) | -0.00 | -0.53 | | | 0.00 | 0.38 | | | -0.04*** | -3.25 | | |
| Household size | number | 6.03 | (2.10) | 0.03*** | 3.30 | | | 0.01** | 2.28 | | | 0.00 | 0.22 | | |
| Dependency ratio | percent | 93.53 | (77.18) | -0.00** | -2.24 | | | -0.00 | -0.08 | | | 0.00 | 1.40 | | |
| Total land owned | ln(ha) | 1.01 | (0.62) | -0.03 | -0.97 | | | 0.03 | 1.36 | | | 0.14* | 1.80 | | |
| Sub-urban area | yes = 1 | 0.27 | | -0.20** | -2.50 | -0.19*** | -2.65 | 0.02 | 0.50 | 0.01 | 0.33 | 0.20 | 0.82 | 0.19 | 0.77 |
| Urban (Addis) area | yes = 1 | 0.10 | | -0.34* | -1.77 | -0.27 | -1.62 | -0.02 | -0.17 | -0.14 | -1.38 | 1.18* | 1.99 | 1.15* | 1.80 |
| Constant | | | | | | | | | | | | 2.34*** | 3.93 | 2.24*** | 3.59 |
| R2/Pseudo R2 for probit | | | | 0.03 | | 0.03 | | 0.10 | | 0.07 | | 0.14 | | 0.10 | |
| Observations | | | | 838 | | 955 | | 838 | | 955 | | 803 | | 918 | |
| | | | | LR (10) = 34.4 | | LR (5) = 42.11 | | LR (10) = 61.90 | | LR(5) = 43.42 | | F(10, 792) = 12.4 | | F(5, 912) = 19.30 | |
| | | | | Prob >chi2 = 0.000 | | Prob >chi2 = 0.000 | | Prob >chi2 = 0.000 | | Prob >chi2 = 0.000 | | Prob> F = 0.000 | | Prob> F = 0.000 | |

Standard errors are robust and adjusted for clusters at the kebele level.

*, **, and *** significant at 10, 5 and 1 percent level, respectively.

Source: Authors' calculations

more. The results show that the share of cross-bred cows on a farm that has more than two cows is 73 percent higher than in the case of a farm that has one or two cows, ceteris paribus. The differences are even more stark when we compare to large farms with 25 cows or more (293 percent). Medium-size farms are also more likely to use purchased wheat bran than the smallest farms, while this is not the case for the large farms. They often rely relatively more on commercial pre-mixes than any other sized farm (22 percent more likely than small farms). Extension agents are also much more used by medium and large farms than small farms, ceteris paribus. Medium farms have also more access to animal health workers. We note however no association between farm size and travel cost to obtain medicines, ceteris paribus.

**Table 9. Associates of adoption of modern dairy practices.**

| Variables | Unit | Mean | standard deviation | Share of cross-bred cow: Tobit | | | | Use of wheat bran: Probit (Average marginal effect) | | | | Use of commercial pre-mix: Probit (Average marginal effect) | | | |
|---|---|---|---|---|---|---|---|---|---|---|---|---|---|---|---|
| | | | | (1) | | (2) | | (3) | | (4) | | (5) | | (6) | |
| | | | | Coeffi-cient | t-value | Coeffi-cient | t-value | Coeffi-cient | t-value | Coeffi-cient | t-value | Coeffi-cient | t-value | Coeffi-cient | t-value |
| Dependent variables | mean | | | 40.63 | | | | 0.40 | | | | 0.09 | | | |
| | (std. dev.) | | | (44.90) | | | | | | | | | | | |
| *Independent variables* | | | | | | | | | | | | | | | |
| Transport cost to Addis | ln(Birr +1) | 3.40 | (1.34) | -115.03** | -2.33 | -118.81** | -2.28 | -0.23*** | -3.60 | -0.24*** | -3.67 | -0.07** | -2.09 | -0.10** | -2.01 |
| Farm size (Default: small farm) | | | | | | | | | | | | | | | |
| Medium farm (3–24 cows) | yes = 1 | 0.47 | | 73.16*** | 3.02 | 69.38*** | 3.07 | 0.12*** | 3.20 | 0.11*** | 2.92 | 0.04*** | 2.99 | 0.07*** | 3.34 |
| Large farm (≥25 cows) | yes = 1 | 0.01 | | | | 293.58*** | 3.91 | | | 0.20 | 1.05 | | | 0.22*** | 4.79 |
| Gender of household head | male = 1 | 0.92 | | 23.78 | 0.63 | | | 0.09 | | | | -0.01 | -0.59 | | |
| Age of household head | number | 48.93 | (13.70) | 0.70 | 1.25 | | | 0.62 | | | | 0.00*** | 2.77 | | |
| Education of household head | number | 3.68 | (4.27) | 6.03** | 2.35 | | | 3.36 | | | | 0.01*** | 6.44 | | |
| Household size | number | 6.03 | (2.10) | -8.12* | -1.86 | | | -0.79 | | | | 0.00 | 0.31 | | |
| Dependency ratio | percent | 93.53 | (77.18) | -0.13 | -1.09 | | | -3.32 | | | | -0.00 | -0.69 | | |
| Total land owned | ln(ha) | 1.01 | (0.62) | -58.05*** | -2.83 | | | -2.67 | | | | -0.05*** | -3.29 | | |
| Sub-urban area | yes = 1 | 0.27 | | -118.29 | -1.53 | -101.76 | -1.34 | -0.80 | -0.09 | -0.76 | | -0.04 | -0.93 | -0.04 | -0.59 |
| Urban (Addis) area | yes = 1 | 0.10 | | -249.00 | -1.25 | -169.50 | -0.84 | -1.10 | -0.39 | -1.33 | | -0.21 | -1.64 | -0.23 | -1.16 |
| Constant | | | | 447.14** | 2.10 | 415.62* | 1.93 | | | | | | | | |
| R2/Pseudo R2 for probit | | | | 0.06 | | 0.09 | | 0.21 | | 0.21 | | 0.37 | | 0.27 | |
| Observations | | | | 804 | | 921 | | 838 | | 955 | | 838 | | 955 | |
| | | | | F(10, 795) = 6.12 | | F(5, 795) = 13.01 | | LR (10) = 227.95 | | LR(5) = 269.08 | | LR(10)) = 141.36 | | LR(5) = 159.92 | |
| | | | | Prob> F = 0.000 | | Prob> F = 0.000 | | Prob >chi2 = 0.0000 | | Prob >chi2 = 0.0000 | | Prob >chi2 = 0.0000 | | Prob >chi2 = 0.0000 | |

Standard errors are robust and adjusted for clusters at the kebele level.

*, **, and *** significant at 10, 5 and 1 percent level, respectively.

Source: Authors' calculations

Third, household characteristics matter for adoption decisions. More educated households are more likely to adopt cross-bred cows and use purchased wheat bran and commercial pre-mixes. This confirms results seen in other studies on adoption of improved practices in agriculture [27]. Land ownership is associated with lower adoption rates of these practices. This is likely because feeds, in this case, can come from the farmers' land, and they do not need to rely on commercial markets to access them.

**Milk productivity analysis.** To understand how the increasing adoption of improved technologies contribute to higher milk productivity, we present results of a milk production function. Detailed data were collected for up to a maximum of 6 cows per surveyed farmer on

**Table 10. Determinants of milk yields, liters per day per cow.**

| Variables | Unit | Mean | Standard deviation | (1) Coefficient | (1) t-value | (2) Coefficient | (2) t-value | (3) Coefficient | (3) t-value | (4) Coefficient | (4) t-value | (5) Coefficient | (5) t-value |
|---|---|---|---|---|---|---|---|---|---|---|---|---|---|
| *Dependent variable* | | | | | | | | | | | | | |
| Milk yield | l/day/cow | 4.43 | 5.00 | | | | | | | | | | |
| *Independent variables#* | | | | | | | | | | | | | |
| Transport costs Addis Ababa | ln(Birr+1) | 3.26 | 1.45 | -2.18*** | -3.21 | | | | | | | | |
| Farm size (default: small farm) | | | | | | | | | | | | | |
| Medium farm (3–24 cows) | yes = 1 | 0.62 | | 1.02*** | 3.57 | | | | | | | | |
| Large farm (≥25 cows) | yes = 1 | 0.02 | | 7.52** | 2.36 | | | | | | | | |
| Cross-bred cow | yes = 1 | | | | | 5.27*** | 9.51 | 3.55*** | 9.20 | 3.23*** | 6.62 | 2.85*** | 8.27 |
| Months since birth calf | ln(months +1) | 1.91 | 0.54 | | | -1.21*** | -4.04 | -0.96*** | -3.70 | -1.00*** | -4.24 | -0.96*** | -4.25 |
| Age cow | years | 6.82 | 2.47 | | | | | -0.74*** | -3.85 | | | -0.37 | -1.99 |
| Age cow squared | years | | | | | | | 0.03*** | 3.09 | | | 0.02 | 1.8 |
| Number of calvings | number | 2.67 | 1.55 | | | | | 1.05*** | 4.07 | | | 0.57** | 2.30 |
| Number of calvings squared | number | | | | | | | -0.08** | -2.66 | | | -0.04 | 1.56 |
| Dry months before calving | number | 6.05 | 3.50 | | | | | -0.06 | -1.30 | | | -0.03 | -0.64 |
| Feed oilseed | yes = 1 | 0.32 | | | | | | 0.29 | 0.86 | | | -0.44 | -0.43 |
| Feed wheat bran | yes = 1 | 0.44 | | | | | | 2.34*** | 5.68 | | | 3.71*** | 3.07 |
| Feed pre-mix | yes = 1 | 0.12 | | | | | | 2.79*** | 2.71 | | | 6.99* | 1.82 |
| Intercept | | | | 11.41*** | 3.69 | 5.08*** | 2.93 | 5.87*** | 3.85 | 4.84*** | 10.87 | 3.31*** | 3.42 |
| Urban/suburban dummies | | | | yes | | no | | no | | no | | no | |
| Woreda dummies# | | | | no | | yes | | yes | | no | | no | |
| Household dummies (fixed effect model) | | | | no | | no | | no | | yes | | yes | |
| Number of observations | | | | 1808 | | 1845 | | 1823 | | 1845 | | 1823 | |
| Number of groups | | | | | | | | | | 954 | | 947 | |
| R-square overall | | | | 0.25 | | 0.47 | | 0.56 | | 0.42 | | 0.49 | |
| R-square within | | | | | | | | | | 0.18 | | 0.25 | |
| R-square between | | | | | | | | | | 0.45 | | 0.52 | |

Robust standard errors are adjusted for clusters at the kebele level

*, **, and *** significant at 10, 5 and 1 percent level, respectively.

Source: Authors' calculations

their milk yield, feeding practices, and cow characteristics. We use in our analysis as the dependent variable the yield of the cow at the time of the survey. Milk yields might be influenced by a number of other factors, so we analyze them in a multi-regression framework. For consistency, we first show associations of milk yields with travel cost and size of the farm (Table 10, specification 1). Both factors are shown to be significantly associated with yields. A doubling of the travel cost to Addis Ababa leads to a reduction in yields of 2 liters/day/cow. Medium and large farms have per cow productivity that is 1.0 and 7.5 liters/day higher, respectively, than what is observed on the smallest farms with one or two cows.

In further regressions, we present results of pooled (specifications 2 and 3) and fixed effect (at the household) (specifications 4 and 5) and parsimonious and Cobb-Douglas production functions for milk. In the production function set-up, we use as right-hand variables breed, age of the cow, dry months between last lactating period and birth, number of months after birth, and feeding practices. In the pooled regression, we further include dummies for each woreda to control for dairy production geography and other explanatory variables associated with woreda characteristics. The computed standard errors are in these specifications clustered at the kebele level.

The results of the pooled models show in both specifications that cross-bred cows are significantly more productive than local cows. Yields are 5.3 l/day/cow higher in the parsimonious model. However, part of the higher yields is explained by better feed that is generally given to those cross-bred cows. Coefficients for cross-bred cows are substantially lower when we control for several other possible explanatory variables. It is especially the feeding of wheat bran and of pre-mixed feed that is showing consistently significant and positive effects on milk yields. In the case of the preferred household fixed effects model, the feeding of wheat bran or commercial pre-mixes to cows leads to an increase of 3.7 and 7.0 l/day/cow, respectively. These results illustrate the higher yield benefits linked to the use of commercial pre-mixed brands, though at higher cost.

Differences are also explained by household specific factors–as shown by the significantly lower yield differences for cross-bred cows in the fixed effect models–as well as the age of the cow. The number of calves that the cow gave birth to also is associated with higher yields. Finally, the longer the period after the cow has given birth, the lower the milk yield. A doubling of the number of months after the birth of the calf leads to a reduction of yields by about 1 liter/day.

Overall, we find that there have been improvements in the last decade in the adoption of improved dairy practices, such as improved breeds, feeding practices, and animal health care, and access to dairy-related livestock services. Improved practices have been adopted especially in the least remote areas and by larger farmers. We also find that the adoption of these improved practices–in particular, the adoption of cross-bred cows and improved feeding practices–are associated with significantly higher milk productivity.

## Conclusions

Relying on a unique combination of primary and secondary data sources, we study recent dynamics in the dairy sector supplying Addis Ababa, the capital of Ethiopia. The city is expanding quickly, and incomes of these urban residents are rapidly increasing as well. This makes the city and its rural hinterland one of the most dynamic regions in the country with important implications for the dairy value chain. We find important changes over time. Upstream, we note increasing spread of supplying regions of liquid milk. In the well-connected regions, we see higher adoption of cross-bred cows and commercial feeds and better access to animal health and dairy-related extension services, contributing to higher milk yields per dairy cows. In more remote areas, we see stagnation–or even a decline–in milk yields. Midstream, the number of processing firms has tripled over the last 10 years, while downstream, an increase in the consumption of liquid milk–especially by the better-off population–is noted. We further document a process of upscaling with larger dairy farms becoming more prevalent. These larger farms are characterized by much higher productivity per cow and per unit of labor. This will likely give further incentives for these types of farms to grow in the future. We also note the large importance of urban farms, mostly focusing on raw liquid milk market sales. Overall, we see important dynamics and the increasing city demand is increasingly providing incentives for transformation of the dairy sector overall.

While the sector has shown important changes, important challenges remain. First, adoption of cross-bred cows has increased, but it is often only the relatively richer households that can afford to invest in the costly but also riskier cross-bred cows. While A.I. is increasingly adopted, the A.I. supply chain is still problematic. The numbers presented in the paper indicate that more effort is further needed around A.I. Second, feed markets are rapidly developing, and an increasing number of dairy farmers are relying on them. However, feed-blending is often done without scientific advice. Given the large returns to milk yields associated with the use of commercial pre-mixes, creating an enabling environment for formal feeding firms to flourish seems needed. Third, urban dairy farms are important actors in dairy value chain. However, it seems that more attention should be paid to this type of farming given the important environmental and health considerations of dairy farming in urban settings. Fourth, the livestock sector is increasingly being questioned for its environmental problems. To reduce enteric methane emissions from the dairy herds, productivity per animal would need to grow and the number of animals to decline. To achieve that vision, it seems that more effort is needed to further stimulate productivity growth through improved input use leading to higher productivity levels in the sector. More effort is also needed to include remote and smaller farms in these changes in Ethiopia's dairy sector.

## Acknowledgments

The authors would like to thank Todd Benson for very useful comments and edits. They further would like to acknowledge Senne Vandevelde and Getachew Abegaz for useful discussions and help with the set-up of the survey.

## Author Contributions

**Conceptualization:** Bart Minten, Seneshaw Tamru, Agajie Tesfaye.

**Formal analysis:** Bart Minten, Yetimwork Habte, Seneshaw Tamru.

**Funding acquisition:** Bart Minten.

**Methodology:** Bart Minten, Seneshaw Tamru.

**Project administration:** Bart Minten.

**Supervision:** Bart Minten, Seneshaw Tamru, Agajie Tesfaye.

**Writing – original draft:** Bart Minten, Seneshaw Tamru.

**Writing – review & editing:** Bart Minten.

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
