## [Decision Letter · Decision Letter 0]

7 May 2020

PONE-D-20-07044

The Transforming Dairy Sector in Developing Countries: Evidence from Ethiopia

PLOS ONE

Dear Dr. Minten,

Thank you for submitting your manuscript to PLOS ONE. After careful consideration, we feel that it has merit but does not fully meet PLOS ONE’s publication criteria as it currently stands. Therefore, we invite you to submit a revised version of the manuscript that addresses the points raised during the review process.

We would appreciate receiving your revised manuscript by Jun 21 2020 11:59PM. To enhance the reproducibility of your results, we recommend that if applicable you deposit your laboratory protocols in protocols.io, where a protocol can be assigned its own identifier (DOI) such that it can be cited independently in the future. For instructions see: http://journals.plos.org/plosone/s/submission-guidelines#loc-laboratory-protocols

We look forward to receiving your revised manuscript.

Kind regards,

Yacob Zereyesus, Ph.D.

Academic Editor

PLOS ONE

Journal Requirements:

3.  Thank you for including your ethics statement: 

'IFPRI's Institutional Review Board

IRB #00007490

FWA #00005121

Writtem consent'

Additional Editor Comments (if provided):

Please see comments from reviewers.

Reviewers' comments:

Reviewer's Responses to Questions

**Comments to the Author**

1. Is the manuscript technically sound, and do the data support the conclusions?

Reviewer #1: Yes

Reviewer #2: Yes

2. Has the statistical analysis been performed appropriately and rigorously? 

Reviewer #1: Yes

Reviewer #2: Yes

3. Have the authors made all data underlying the findings in their manuscript fully available?

Reviewer #1: Yes

Reviewer #2: Yes

4. Is the manuscript presented in an intelligible fashion and written in standard English?

Reviewer #1: Yes

Reviewer #2: Yes

5. Review Comments to the Author

Reviewer #1: This study gives in-depth analysis of structural transformation in the Ethiopian dairy sector, focusing on Addis Ababa, the capital of Ethiopia. The study found that dairy consumption in the metropolitan area of Addis Ababa has increase between 2005 and 2016 but still relatively low compare to average global consumption. Milk yield has been increasing in the Urban area over the decade while rural farmers have been experiencing almost stagnant milk yield over the decade. One of the main determinants of productivity gain in the urban farms is the adoption of new breeds of cow.

I have read the paper very well and I think it is suitable for publication in this journal. Below are my comments.

Major:

1. The authors should briefly summarize their finding as part of the introduction. It is general practice now for introduction to include the summary of important findings in an article.

2. The authors run Probit and Tobit modesl: I don't know why they decided not to provide description of their model and variables in the paper, including assumptions made. A brief description of how their model is set up is necessary.

Minor:

1. There should be at least a footnote defining "Woreda" the first time it is mentioned on page 4.

2. Similarly, "kebeles" should be described briefly as it is first mentioned in page 5.

3. In table 1, under the "Characteristics of dairy farm households", the authors should provide information about the distribution of the variables. Something like standard deviation in parenthesis will be helpful.

4. Inconsistency in units labeling: In table 3; are mean and median units? The unit labeling should be consistence.

5. Table 5: "Travel time to access an A.I. agent, in minutes" - the unit is not consistent with the rest. The unit should be "Minutes" while variable should be labeled "Median travel time to access an A.I. agent."

6. Similarly, the unit under "Administrative data" should be corrected

7. On page 16: "livestock death rates declined over the last decade"; declined from what value to what value?

Reviewer #2: Manuscript Number: PONE-D-20-07044

Title of the article: The Transforming Dairy Sector in Developing Countries: Evidence from Ethiopia

Corresponding author: Bart Minten, International Food Policy Research Institute, Washington DC, USA.

Reviewer: Pacem Kotchofa, International Livestock Research Institute, Nairobi, Kenya.

I. Summary of the article

The authors attempt to shed light on the transformation of the Ethiopian's agricultural food systems over the past decades using a wide range of methods and data, meaning, both primary and secondary data sources. Specifically, they focus on the dairy value chain supplying the capital city of the country, i.e., Addis Ababa. Strategically, they emphasize changes that happened from the upstream, midstream, and downstream of the dairy production chain to the final consumption point in Addis.

To communicate this transformation story effectively, authors gathered historical data multiple outcomes such as production, processing firms, cost, income, and consumption expenses related to various animal source foods (ASF) like milk, butter, cheese, and yogurt. This set of outcome data are further disaggregated per income quintiles, i.e., from the poorest to the wealthiest consumer groups. They have also made good use of both visual representations like figures as well as several descriptive tables where they provide the reader with at least two substantial evidence to support their arguments:

- the change over time of the milk yield, i.e., liter/cow/day is disproportionated due to factors such as the location of the farms, breeds, and other inputs like feeds but also

- access to livestock services, which represents another significant factor that underlie this transformation of the dairy value chain and disproportionated spatial productions.

Besides, the authors have made use of various econometric models to strengthen and sharpen the results of their analyses. They estimated several discrete choice models like Probit and Tobit to give insights on the importance of the above-listed factors in explaining the adoption decisions of improved technologies in milk production as well as access to livestock extension services. They have also provided some interesting marginal values to signal predictive probabilities in farmers' adoption decisions, given a change in these explanatory variables. In addition to these discrete choice models, the authors have also investigated the determinants of milk yield function by estimating again various production models, which range from basic OLS to a pooled and fixed effects models as well as the Cobb-Douglas production functions.

After a series of regression tables, figures, and discussions, the authors have concluded that, indeed, essential changes did take place over time around Addis Ababa regarding the production and consumption of dairy products. Downstream of the value chain, most of these dynamic changes are attributed to the rapidly expanding incomes and food preferences of Addis's urban residents. Besides, higher adoption of crossbred cows, commercial feeds, access to animal health, and dairy-related extension services was noticed upstream on the value chain even though these effects are spatially disproportionated. Authors pointed out that in more remote areas, this improving transformation is less noticeable or worse, showing, for instance, at some places serious declining trend in milk yield. The midstream of the chain is marked by a significant increase in the number of processing firms which tripled over the last ten years.

Overall, this manuscript did an excellent job demonstrating that substantial changes happened over the past decades in the dairy value chain system, supplying the capital city of Ethiopia, Addis Ababa. However, the reviewer did also notice some keys points that the authors might need to address before the publication of this manuscript. Here, the reviewer provided their comments classified into major and minor concerns as follows.

II. Major concerns

The authors are off to a good start with this manuscript. However, there are a few points that the reviewer deems critical to be addressed before the publication of this piece of work.

1- This study requires at least one additional section where the authors might provide their recommendations or course of action moving forward in terms of policy recommendations. The manuscript currently appears as a fascinating study of past events without any tangible directives looking into the next ten years or so. For instance, such a section could include policy recommendations that might target those farmers in rural and remote areas that seem not to be taking advantage of the overall positive transformation happening in the dairy value chain supplying Addis. Hence, designing policy interventions that enable them to have access to livestock services and support them in the adoption of improved technologies could enhance those rural-urban linkages that the authors are currently trying to capture through the upstream to downstream value chain analysis. Also, based on the other exciting results the authors have already found, they have the luxury to be even more creative in writing such a section to capture the attention of their readers better.

2- Here goes a section of the manuscript, on page 8, that raises serious concerns "While the official population figure for Addis Ababa was 3.27 million in 2015, this is considered an underestimate of the population of Addis Ababa metropolis, given increasing urbanization in areas surrounding the city." The reviewer is interested in knowing the reason why the authors seem doubting these national statistics while using other data from the same source to support their argument? Keeping this sentence at its current state might appears that they select whichever data favor their story.

3- Authors have estimated several models, i.e., discrete choice models and production function models. It will be easier for the reader to follow the interpretation of their results, knowing the mathematical functional forms of the two sets of regressions they estimated. Also, once the mathematical forms specified, the information in the regression tables will be better structured as these tables will now only report estimated parameters already explained in the models.

4- Interpretations of the results in Table 8 and Table 9: Given that the variable "Transport cost to Addis Ababa" is specified in a logarithmic form, i.e., ln(cost), the reviewer suggests rewriting the interpretations correctly using a percentage change in the explanatory variable instead of "a double of the cost." Also, for easy reading, it will be better to mention which regression tables (8 or 9) results are being interpreted and discussed.

Results in Table 8: The authors need to provide the precision on whether the probit marginal effects presented are (i) Marginal effect at the mean (MEM); (ii) Average marginal effect (AME); or (iii) Marginal effects at representative values (MER) or others.

Also, the reviewer would suggest using "probability" when interpreting marginal effects instead of "likelihood" as marginal effects give the predicted probability of an outcome variable given a specific change.

For instance, here is a section of the manuscript on Page 23: "A doubling in the distance reduces the likelihood of using purchased wheat bran by 23 to 24 percent and of using commercial pre-mixes by 7 to 10 percent, depending on the specification". The reviewer thinks that distance is fixed; only the cost of transportation varies here. Thus, the authors might consider rephrasing not only the sentence but also the interpretation of the change. An increase in the cost of transportation does not necessarily imply that the actual distance has increased. Also, the variable used here is transportation cost and not distance in km.

5- Questions about the results in Table 8: How do authors interpret the value of the mean of household education, i.e., 3.68 years? Does the education variable refer to the number of years in general schooling or years of technical agricultural knowledge?

6- Questions about the presentation of results in Table 10: The reviewer is interested in understanding the reason why authors added significant levels stars to the t-values instead of the actual coefficients estimated?

III. Minor problems

While this study appears to be sound and thoughtful, the reviewer has made a list of a few minor suggestions and questions that could help authors in improving the readability and the flow of their manuscript.

1- The wording at some points of the manuscript is unclear, making it sometimes difficult to follow as sentences becoming longer and longer. The reviewer would advise authors to improve the flow and readability of their manuscripts, especially around the interpretations and discussion sections, but rewriting some of the long sentences.

2- This might be a personal preference, but the reviewer wonders why the authors included "developing countries" in their study title, knowing that they have studied just one developing country. Also, the study did not technically cover that entire country either. It will be more focused and concise to have something like "The Transforming Dairy Sector in Ethiopia."

3- It will be interesting for the authors to provide a map of the country, showing the area of study and some representation of the value chains to attract the attention of their readers.

4- Referring to Bennet's law' could strengthen the theoretical foundation of the authors' arguments about the historical transformation of the dairy value chain, at least, from the demand side. Bennet's law states that as income increases, not only does the quantity of food increase less than proportionally but also the composition of the food basket changes. Specifically, the consumption of starchy staple food declines as income rises. The authors could use this framework to support change in food preference and demand for more ASF as income expand rapidly among urban residents of Addis.

5- Consistency with the writing style of quantitative information will be helpful. For instance, here is a section of the manuscript on page 4: "Ninety-seven dairy farming households were interviewed in Addis Ababa, 256 in suburban areas in the Oromia Special Zone surrounding Finfinne, and 602 in rural areas."

6- Authors should consider providing at least a footnote explaining what is regarded as woredas and kebele in this study.

7- The authors should also consider providing proper titles for each figure in the manuscript. I addition, some figures like 1,3,5, and 7 do not have variable names on their X axes.

8- Reformat Tables 8 and 9: Make sure that the formation contained in one column is not found under another column if they are supposed to provide different data. For instance, in Table 9 in column (3), the expression "Use of wheat bran Probit (marginal effect)" span under column (4).

9- It will also be meaningful if the authors could provide the likelihood ratios chi-square of the discrete choice models estimated to determine the goodness of fit of the models.

10- Consistent in the presentation of the regression tables: it will be helpful if the authors could adopt one formatting style. In previous tables, i.e., 8 and 9, the SD was in brackets under the estimated coefficients. In contrast, in Table 10, they are listed under a different column along with t-values and the estimated coefficients.

6. PLOS authors have the option to publish the peer review history of their article (what does this mean?). If published, this will include your full peer review and any attached files.

Reviewer #1: No

Reviewer #2: Yes: Pacem Kotchofa

---

## [Author Response · Author response to Decision Letter 0]

11 Jun 2020

We would like to thank the reviewers for their insightful comments. We believe that by bringing in these changes in the paper, the quality of the paper has increased significantly. Please find below how we have addressed the different suggestions. We give a point-by-point response to the different comments and concerns. 

Comments from Reviewer 1

Major comments:

1. The authors should briefly summarize their finding as part of the introduction. It is general practice now for introduction to include the summary of important findings in an article.

Response: We have now included a summary of the findings in the introduction. 

2. The authors run Probit and Tobit modes: I don't know why they decided not to provide description of their model and variables in the paper, including assumptions made. A brief description of how their model is set up is necessary.

Response: We have added a brief description in the methods section on the choice of models in the paper.

Minor:

1. There should be at least a footnote defining "Woreda" the first time it is mentioned on page 4.

Response: We agree with this comment. Footnotes are not permitted as per the guidelines of PLOS ONE. We have therefore given its definition in brackets after it is first used. 

2. Similarly, "kebeles" should be described briefly as it is first mentioned in page 5.

Response: Same as above. As footnotes are not allowed as per the guideline, we added the definition in brackets.

3. In table 1, under the "Characteristics of dairy farm households", the authors should provide information about the distribution of the variables. Something like standard deviation in parenthesis will be helpful.

Response: We included standard deviation in parenthesis. 

4. Inconsistency in units labeling: In table 3; are mean and median units? The unit labeling should be consistence.

Response: Thanks. We incorporated the appropriate unit of measurements. 

5. Table 5: "Travel time to access an A.I. agent, in minutes" - the unit is not consistent with the rest. The unit should be "Minutes" while variable should be labeled "Median travel time to access an A.I. agent."

Response: Thanks. We corrected accordingly. 

6. Similarly, the unit under "Administrative data" should be corrected

Response: Thanks. We adjusted accordingly.

7. On page 16: "livestock death rates declined over the last decade"; declined from what value to what value?

Response: We have included the exact value on the decline in the livestock death rate over the last decade.

Comments from Reviewer 2: 

I. Major concerns

1- This study requires at least one additional section where the authors might provide their recommendations or course of action moving forward in terms of policy recommendations. The manuscript currently appears as a fascinating study of past events without any tangible directives looking into the next ten years or so. For instance, such a section could include policy recommendations that might target those farmers in rural and remote areas that seem not to be taking advantage of the overall positive transformation happening in the dairy value chain supplying Addis. Hence, designing policy interventions that enable them to have access to livestock services and support them in the adoption of improved technologies could enhance those rural-urban linkages that the authors are currently trying to capture through the upstream to downstream value chain analysis. Also, based on the other exciting results the authors have already found, they have the luxury to be even more creative in writing such a section to capture the attention of their readers better.

Response: Thanks. A final section with policy recommendations has been added.

2- Here goes a section of the manuscript, on page 8, that raises serious concerns "While the official population figure for Addis Ababa was 3.27 million in 2015, this is considered an underestimate of the population of Addis Ababa metropolis, given increasing urbanization in areas surrounding the city." The reviewer is interested in knowing the reason why the authors seem doubting these national statistics while using other data from the same source to support their argument? Keeping this sentence at its current state might appears that they select whichever data favor their story.

Response: Good point. That sentence has now been dropped and we only talk about estimates of the metropolitan area.

3- Authors have estimated several models, i.e., discrete choice models and production function models. It will be easier for the reader to follow the interpretation of their results, knowing the mathematical functional forms of the two sets of regressions they estimated. Also, once the mathematical forms specified, the information in the regression tables will be better structured as these tables will now only report estimated parameters already explained in the models.

Response: An expanded methodology section is now included.

4- Interpretations of the results in Table 8 and Table 9: Given that the variable "Transport cost to Addis Ababa" is specified in a logarithmic form, i.e., ln(cost), the reviewer suggests rewriting the interpretations correctly using a percentage change in the explanatory variable instead of "a double of the cost." Also, for easy reading, it will be better to mention which regression tables (8 or 9) results are being interpreted and discussed.

Response: We have rewritten the section taking these valuable recommendations in to consideration.

Results in Table 8: The authors need to provide the precision on whether the probit marginal effects presented are (i) Marginal effect at the mean (MEM); (ii) Average marginal effect (AME); or (iii) Marginal effects at representative values (MER) or others.

Also, the reviewer would suggest using "probability" when interpreting marginal effects instead of "likelihood" as marginal effects give the predicted probability of an outcome variable given a specific change. For instance, here is a section of the manuscript on Page 23: "A doubling in the distance reduces the likelihood of using purchased wheat bran by 23 to 24 percent and of using commercial pre-mixes by 7 to 10 percent, depending on the specification". The reviewer thinks that distance is fixed; only the cost of transportation varies here. Thus, the authors might consider rephrasing not only the sentence but also the interpretation of the change. An increase in the cost of transportation does not necessarily imply that the actual distance has increased. Also, the variable used here is transportation cost and not distance in km.

Response: We estimated the average marginal effect and have made this now clear in the title of the table. We also substituted probability for likelihood in our interpretation of results. Distance has been replaced everywhere by travel cost.

5- Questions about the results in Table 8: How do authors interpret the value of the mean of household education, i.e., 3.68 years? Does the education variable refer to the number of years in general schooling or years of technical agricultural knowledge?

Response: It refer to the average number of years in general schooling.

6- Questions about the presentation of results in Table 10: The reviewer is interested in understanding the reason why authors added significant levels stars to the t-values instead of the actual coefficients estimated?

Response: We have changed and have put the significance levels next to the coefficients. 

III. Minor problems

1- The wording at some points of the manuscript is unclear, making it sometimes difficult to follow as sentences becoming longer and longer. The reviewer would advise authors to improve the flow and readability of their manuscripts, especially around the interpretations and discussion sections, but rewriting some of the long sentences.

Response: We have edited the whole manuscript as to improve readability.

2- This might be a personal preference, but the reviewer wonders why the authors included "developing countries" in their study title, knowing that they have studied just one developing country. Also, the study did not technically cover that entire country either. It will be more focused and concise to have something like "The Transforming Dairy Sector in Ethiopia."

Response: We have changed the title accordingly.

3- It will be interesting for the authors to provide a map of the country, showing the area of study and some representation of the value chains to attract the attention of their readers.

Response: We had a map drawn using Arc GIS (a licensed software) illustrating the location of the surveyed households. However, PLOS ONE cannot publish any images created using copyrighted software. Thus, we are forced to drop it.

4- Referring to Bennet's law' could strengthen the theoretical foundation of the authors' arguments about the historical transformation of the dairy value chain, at least, from the demand side. Bennet's law states that as income increases, not only does the quantity of food increase less than proportionally but also the composition of the food basket changes. Specifically, the consumption of starchy staple food declines as income rises. The authors could use this framework to support change in food preference and demand for more ASF as income expand rapidly among urban residents of Addis.

Response: Thanks. That reference to Bennet’s law has now been made.

5- Consistency with the writing style of quantitative information will be helpful. For instance, here is a section of the manuscript on page 4: "Ninety-seven dairy farming households were interviewed in Addis Ababa, 256 in suburban areas in the Oromia Special Zone surrounding Finfinne, and 602 in rural areas."

Response: We have checked for consistency of presenting quantitative information throughout the manuscript and modified accordingly. 

6- Authors should consider providing at least a footnote explaining what is regarded as woredas and kebele in this study.

Response: As footnotes are not permitted as per the guideline of PLOS ONE, we have given definitions in brackets for woredas and kebeles.

7- The authors should also consider providing proper titles for each figure in the manuscript. I addition, some figures like 1,3,5, and 7 do not have variable names on their X axes.

Response: The graphs have been adjusted accordingly. We have included titles in the text as well.

8- Reformat Tables 8 and 9: Make sure that the formation contained in one column is not found under another column if they are supposed to provide different data. For instance, in Table 9 in column (3), the expression "Use of wheat bran Probit (marginal effect)" span under column (4).

Response: Each expression represents two columns for the short and the long form of the results. Therefore, in Table 9 the expression "Use of wheat bran Probit (marginal effect)" spans from column 3 to 4. To make this clearer, we have now moved the column numbers under model title(expression) so as to avoid confusion. 

9- It will also be meaningful if the authors could provide the likelihood ratios chi-square of the discrete choice models estimated to determine the goodness of fit of the models.

Response: We have now incorporated the likelihood ratios chi-square in the tables. 

10- Consistent in the presentation of the regression tables: it will be helpful if the authors could adopt one formatting style. In previous tables, i.e., 8 and 9, the SD was in brackets under the estimated coefficients. In contrast, in Table 10, they are listed under a different column along with t-values and the estimated coefficients.

Response: To ensure consistency, we reformatted table 8 and 9 and presented t-values (instead of standard error) in a separate column next to the estimated coefficients.

---

## [Decision Letter · Decision Letter 1]

28 Jul 2020

The Transforming Dairy Sector in Ethiopia

PONE-D-20-07044R1

Dear Dr. Minten,

We’re pleased to inform you that your manuscript has been judged scientifically suitable for publication and will be formally accepted for publication once it meets all outstanding technical requirements.

Kind regards,

Yacob Zereyesus, Ph.D.

Academic Editor

PLOS ONE

Additional Editor Comments (optional):

Reviewers' comments:

Reviewer's Responses to Questions

**Comments to the Author**

1. If the authors have adequately addressed your comments raised in a previous round of review and you feel that this manuscript is now acceptable for publication, you may indicate that here to bypass the “Comments to the Author” section, enter your conflict of interest statement in the “Confidential to Editor” section, and submit your "Accept" recommendation.

Reviewer #1: All comments have been addressed

Reviewer #2: All comments have been addressed

2. Is the manuscript technically sound, and do the data support the conclusions?

Reviewer #1: Yes

Reviewer #2: Yes

3. Has the statistical analysis been performed appropriately and rigorously? 

Reviewer #1: Yes

Reviewer #2: Yes

4. Have the authors made all data underlying the findings in their manuscript fully available?

Reviewer #1: Yes

Reviewer #2: Yes

5. Is the manuscript presented in an intelligible fashion and written in standard English?

Reviewer #1: Yes

Reviewer #2: Yes

6. Review Comments to the Author

Reviewer #1: (No Response)

Reviewer #2: Dr. Bart Minten et al.,

Let me first of all, thank you all for the effort in addressing all my questions and suggestions. I also much appreciate your feedback and genuinely hope that they have indeed increased the quality of the paper significantly. I look forward to reading the final version of this paper, which I still find exciting. Hence, I wish you all great success with the manuscript and future research projects.

Best regards,

Pacem Kotchofa

7. PLOS authors have the option to publish the peer review history of their article (what does this mean?). If published, this will include your full peer review and any attached files.

Reviewer #1: No

Reviewer #2: **Yes: **Pacem Kotchofa

---

## [Editor Report · Acceptance letter]

31 Jul 2020

PONE-D-20-07044R1 

The Transforming Dairy Sector in Ethiopia 

Dear Dr. Minten:

I'm pleased to inform you that your manuscript has been deemed suitable for publication in PLOS ONE. Congratulations! Your manuscript is now with our production department. 

Kind regards, 

on behalf of

Dr. Yacob Zereyesus 

Academic Editor

PLOS ONE